**communications** engineering

**Perspective**

# Emerging sensor technologies and physics-guided methods for monitoring automotive lithium-based batteries
Xia Zeng & Maitane Berecibar ✉

As the automotive industry undergoes a major shift to electric propulsion, reliable assessment of battery health and potential safety issues is critical. This review covers advances in sensor technology, from mechanical and gas sensors to ultrasonic imaging techniques that provide insight into the complex structures and dynamics of lithium-ion batteries. In addition, we explore the integration of physics-guided machine learning methods with multi-sensor systems to improve the accuracy of battery modeling and monitoring. Challenges and opportunities in prototyping and scaling these multi-sensor systems are discussed, highlighting both current limitations and future potential. The purpose of this study is to provide a comprehensive overview of the current status, challenges, and future directions of combining sensors with physically guided methods for future vehicle battery management systems.

According to the International Energy Agency, electric car sales neared 14 million in 2023, with battery electric cars accounting for 70% of the stock[1]. Such increasing sales are pushing up demand for innovations across the battery value chain, including large-scale manufacturing, developing new chemistries (e.g., lithium-sulfur, solid-state, and sodium-ion), recycling, etc.[2]. Additionally, these innovations are strongly judged on a series of metrics with an often complex connection between the requirements set at the fundamental research stage and the eventual application. For example, for an electric vehicle (EV), requirements on safety, range, available pack installation space, cost, power, and lifespan will heavily inform requirements at the battery cell level, such as energy density, chemistry, cell design, and calendar and cycle life[3]. To extract the maximum performance from the developed battery technologies, for example, the widely commercially available lithium iron phosphate, lithium nickel manganese cobalt oxide (NMC), and lithium nickel cobalt aluminum oxide batteries, a battery management system (BMS) is used. The BMS is an embedded system that is interconnected with all battery cell-pack components and the control computation unit. It senses parameters, such as voltage, current, and temperature, of batteries and aims to promptly protect the battery against potential abuses and failures (e.g., thermal runaway) by continuously monitoring and controlling key battery indicators, such as state of charge (SoC) and state of health (SoH)[4]. Moreover, a good BMS is expected to be easily implementable across different chemistries and end-user applications. Consequently, choosing proper sensor types and designing robust monitoring and controlling methodologies are of great importance and research interest in BMS development.

Sensors used in the commercial BMS are typically attached to the cell surfaces or tabs, measuring individual cells' overall voltage, current, and temperature. Several types of methods have been developed with these signals to monitor and control the battery. The most straightforward way is empirical methods, which map the states of interest with adequate offline characteristic measurements[5,6]. They are simple and easy to implement, thus widely used in industry. Meanwhile, the growing amount of collected battery data and recent advances in data science provide an opportunity to incorporate sophisticated machine learning (ML) methods into BMS innovations, for example, in SoC monitoring[7], lifetime prediction[8–10], lithium plating detection[11], degradation pattern identification[12], and fast charging[13–15], etc. ML-based methods are mechanism-free and do not require an explicit mathematical model to describe the nonlinear coupled dynamics of batteries. However, the accuracy and robustness of the methods mentioned above have often met with limited success due to their high data quality and quantity requirements, lack of generalizability to out-of-sample scenarios, and inability to produce physically consistent results[16]. Therefore, models and methods involving battery principles are needed.

Rechargeable batteries function based on the movement of ions between the electrodes. As usage patterns change, the battery material properties and performance gradually degrade over the lifetime[17–19]. Fundamentally, the degradation and potential safety issues of lithium-ion batteries (LIBs) arise from various unwanted side reactions within different components of the cells: the electrodes, the electrolyte, and the separate and current collectors[20]. For example, in cases when solid diffusion is slow (e.g., at low temperatures, high SoC, or in materials with high energy barriers) and

Electromobility Research Centre (MOBI), Department of Electrical Engineering and Energy Technology, Vrije Universiteit Brussel, Brussels, Belgium.
✉e-mail: Maitane.Berecibar@vub.be

the current density during charging is too high (e.g., with high charging rates), some lithium ions may deviate from the primary intercalation reaction and become metallic solid lithium. Such a process is known as lithium plating. Theoretically, the plating is reversible as long as electrical contact with the anode is present. However, as cycled, such contact might be lost, resulting in dead lithium covering the anode particle surfaces and reducing the active area. Additionally, some plated lithium metal reacts with the electrolyte and becomes insoluble products, known as the secondary solid electrolyte interphase (SEI)[21]. These side reactions take place continuously, leading to decreased lithium inventory, increased resistance[22], capacity loss[23], and potential safety concerns[24,25].

Understanding these side reactions' mechanisms, interdependencies, and consequences is crucial for more accurate online monitoring and safer control of LIBs. This is particularly important for EV batteries, which are often designed and assembled to be compact, limiting the use of advanced diagnostic tools such as X-ray diffraction, nuclear magnetic resonance, electron paramagnetic resonance, and transmission electron microscopy to analyze battery performance during cycling[19,26]. Consequently, previous research heavily relies on utilizing measured signals (generally the current, terminal voltage, temperature, and impedance) to model the battery multiphysics dynamics and infer the potential degradation or failure mechanisms. For example, variations in the coulombic efficiency, plateaus in the terminal voltage, and features from the differential voltage (DV) curves can all serve as indicators of lithium plating[24]. Additionally, by relating the change of anode porosity with the increase in surface film thickness, irreversible lithium plating can be incorporated into the electrochemical model[27]. Despite extensive studies, these methods are often limited to specific scenarios and assumptions. In particular, DV curves are generally obtained when the operating current remains constant and low. The reported electrochemical models are usually developed by assuming homogeneous materials and may not be suitable for real-time control and estimation without adequate model-order reduction and simplification[25].

These challenges arise because battery dynamics involve simultaneous electrochemical, mechanical, and thermal interactions, for which external sensor signals can provide only limited insights. At the same time, most methods overly depend on them to deconstruct battery aging and failure mechanisms and their inter-dependencies. Consequently, recent years have seen increasing interest in identifying more direct and effective physical-meaning indicators from a broader range of sensor signals, which eventually enables passive monitoring of LIBs internal dynamics[28]. For example, using pressure gauges, acoustic probes, and gas sensors allows measurement of parameters such as cell surface[29] and internal[30] strain, time of flight (ToF)[31–35], as well as the $CO_2$[36] at various locations within a cell. These spatially resolved parameters can be used to estimate battery states[33,34,37] and to investigate and detect aging mechanisms[32,33,35] and thermal runaway[36]. While these technologies show great promise in enhancing our knowledge of macroscopic properties and improving battery safety, lifespan, and sustainability[25,38], they have not been comprehensively compared from a BMS perspective in other literature.

This study aims to summarize recent key progress in emerging sensing techniques used in battery monitoring, including their working principles and representative examples, while highlighting their advantages and disadvantages. Additionally, we provide an overview of physics-guided battery modeling and control methods, discussing the opportunities and challenges of incorporating emerging sensing signals into these frameworks. Lastly, we present the challenges of integrating smart sensing technologies into future BMS. We anticipate this study will inspire future developments in BMS and push forward the practical applications of newly developed vehicle batteries.

## Emerging sensing techniques in battery monitoring
### Battery sensor types, locations, and pre-requirments
Based on their measurement or stimulus, sensors used in the battery domain can be grouped into four categories: (a) electric sensors for voltage and current measurements, (b) thermal sensors for temperature, flux, and specific heat measurements, (c) acoustic and ultrasonic sensors for wave amplitude, phase, and spectrum measurements, and (d) mechanical sensors for pressure, strain, and stress measurements[39]. Additionally, when the battery ages, it may experience unwanted localized temperature distributions and expansions, potentially leading to safety concerns such as overheating or mechanical failures. Inhomogeneous thermal gradients could also trigger localized lithium plating, particularly along the cell edge with anode overhang[40,41]. Hence, monitoring the thermal-mechanical changes across various battery sites facilitates a more precise assessment of their health and performance, enabling the classification of sensors as either attached or embedded/implantable.

Inspired by flexible sensors used in robotics[42], Fig. 1 shows a concept scheme of a pouch cell with attached and injected sensors. Embedded sensor arrays can be placed under cell shields or between electrode layers, using different channel patterns for various types of sensing, such as multi-axial strain and pressure. Special isolating materials would protect the sensing lines from the harsh chemical environment inside the battery, and communication cables are needed to transmit data. Wireless transmission may be preferred to confront the battery environment in terms of chemical reactivity and manufacturing constraints. Attached sensors generally need to be fastened on the cell surface to get effective and reliable measurements, as shown in Fig. 2a and b. This can be achieved by applying external pressure[43–49], glue[50,51], water-based gel[52], and epoxy adhesive[53] on the sensors. While optimal external pressure can potentially extend battery life by reducing lithium loss[54], it requires careful data analysis. To be more specifically, to apply an ideal homogeneous external pressure on the cell surfaces, different fixtures (e.g., springs and plates as shown in Fig. 2b) are needed[45–47], which potentially affect the battery's thermal dynamics. Therefore, it is crucial to consider these effects and find ways to mitigate their impact on the measured parameters and developed methods, especially if significant thermal dynamics are presented in the studied LIBs. As for the injected sensors, especially fiber optical sensors[30,55–57], their development presents additional challenges, which will be further discussed in later sections.

### Acoustic and ultrasonic sensors
In analogy to visible and ultraviolet light, the terms 'sound' and 'ultrasound' describe the propagation of a mechanical perturbation in different frequency ranges. Due to their high sensitivity, versatility, and cost-effectiveness, both sensors have been widely used in non-destructive detection and structure health management in different domains[58,59]. With a LIB, one or two transducers, typically piezoelectric wafer transducers (piezos), are mounted to send and receive signals, as shown in 2a. Piezos are inexpensive, available in very fine thicknesses (0.1 mm for ceramics and 9 $\mu m$ for polymer film), and easily integrated into structures. They operate on the piezoelectric principle, generating an electric charge in response to applied mechanical stress. Thus, they can function as both acoustic actuators and sensors. Lead zirconium titanate ceramics (PZT) and polyvinylidene fluoride (PVDF) are the most commonly available materials. However, due to their weaker inverse piezoelectric properties and high compliance, PVDF-based transducers perform poorly, yet they cannot be embedded in composite structures due to the loss of properties during typical composite curing conditions. Therefore, PZT is the most popular choice for transducer material among structure health management researchers[60]. As presented in 2a, the initial pulse is first generated and sent from the transducer, then passes through each interface of LIBs. Depending on the degree of mismatch in the sound speed $c$ between adjacent layers and whether $c$ increases or decreases from one layer to the next, some fractions of the wave are transmitted while some are reflected. Additionally, the wave is attenuated (e.g., loses energy) as it travels through the bulk region of each layer. As each interface is an opportunity for the pulse to split, the acoustic behavior of the cell quickly becomes complicated as each new wave interacts not only with interfaces (creating even more waves) but also with each other[33,61,62]. Thus, the received signal, as presented in 2a, is an averaged estimation of a fusion dataset of

**Fig. 1 | The concept of a multi-sensor system for a large pouch battery.** [Top] The battery with attached and embedded sensor layers. [Lower left] The layered structure of the cell, with embedded sensor layers under the shields. [Lower right] The design concept of one embedded sensor layer. (For better understanding, readers are encouraged to refer to the colored version of the figure).

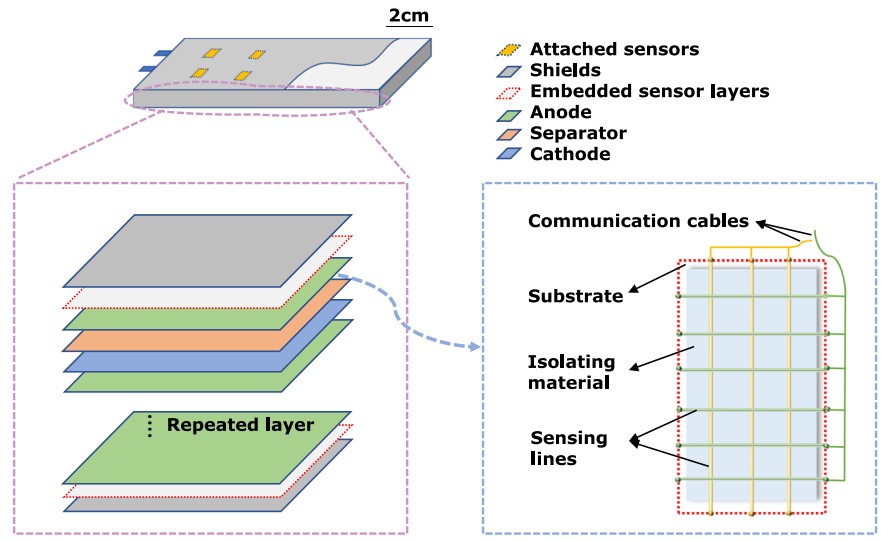

'echo chamber,' with two peaks observed in the signal amplitude (SA) and decreased through the thickness of the cell.

Authors in ref. 44 measured ultrasonic data at 36 locations on a Samsung J5 lithium cobalt oxide (LCO) battery. Results suggest that the magnitude of the first (and also fast) peak decreases while the second (and also slow) peak increases during the charge. These contrasting changes in magnitude indicate a shift in the lithiation state of each electrode. That is, during charge, increased lithiation results in a higher-density anode and a correspondingly lower density of the active material in the cathode; additionally, several studies have shown that the anode contributes more significantly to the overall cell expansion[63,64]; thus, by assuming equivalence between the speed of sound through the material and the velocity of the acoustic wave propagating through the material, it takes more time for the acoustic wave to propagate through the anode than the cathode, resulting in slow and fast peak, respectively. X-ray computed tomography results of the cell further confirmed the hypothesis. Similar experimental results are reported as ToF and SA vary nearly monotonically over a charge-discharge cycle[33,50,51,65,66]. Moreover, with the ambient temperature increasing from -10 to 60 °C, the change of the ToF (ΔToF) of the first echo peak increased linearly, whereas the SA decreased monotonically. Comparable linear dependence presents between ΔToF and C-rates when the depth of discharge is 0 to 75%[65]. As degradation processes, ToF gradually shifts to a smaller nominal value, indicating an increased modulus-to-density ratio and irreversible structure transformation of the aged cells. Furthermore, the ToF and SA at the end of the rest period of every cycle showed a solid nonlinear correlation with SoH[51]. Apart from using copious amounts of experimental data to identify and train empirical functions, extracted features can be directly entered as static states in an extended Kalman filter to estimate battery SoC and SoH simultaneously[67]. The validation results show that the proposed method can converge from poorly defined initial states and track well within five cycles, whereas the conventional approach using only voltage measurements fails without any sign of converging.

The observed variations in acoustic and ultrasonic features stem from the structure and material property differences inside LIBs, making these sensors excellent candidates for revealing and detecting potential degradation and failure mechanisms. The by-products, for example, particle cracks and gas, alter the material density, stiffness, cell thickness, etc, further affecting sound propagation speed and creating distinct features in received signals. As highlighted in Fig. 3a, the acoustic parameters dropped sharply and then disappeared when the cycling mode switched from C/10 constant current (CC) charge/discharge to 1C constant current constant voltage (CCCV) charge and C/10 CC discharge at 5 °C. Such distinctive changes could be related to potential lithium plating, as the cell was cycled at low temperature and a higher C-rate, which, as discussed earlier, are more likely to yield lithium plating. However, such loss of acoustic signal at low temperatures or high C-rates is not necessarily permanent. As reported in ref. 68, loss of acoustic signal is also a strong indicator of gassing, as the transmission efficiency from transducer to air is much lower than that from transducer to liquid or solid due to a greater impedance mismatch ratio. Therefore, using acoustic attenuation alone as a basis for lithium plating is ineffective. Instead of analyzing voltage-dependent cycling protocol data, the author in ref. 32 compared the ToF shift of different charging processes by fixing the charged capacity at the nominal capacity of the studied commercial LIBs. As shown in Fig. 3b, at 10 °C, when the 1C rate fixed-capacity charge finished, the ToF shift was completely different from that of the benchmark C/15 charge, with a significant difference in the endpoints; whereas very less differences in the ToF endpoint were seen for non-plating conditions, for example, 0.5C rate. More interestingly, with many hours of relaxation after the charge, the ToF endpoint of the 1C charge did not downshift, though the cell temperature did. These findings reveal that, though acoustic signals are sensitive to temperature variations, there is a less significant effect on the ToF endpoint difference. The post-mortem analysis results confirmed that the cells with the most amount of lithium plated tend to have higher ToF endpoint differences. The above-discussed studies rely on single- or multiple-located transducers to measure specific cell regions, though lithium plating can happen in different cell locations. To investigate the entire cross-section plating behaviors, authors in ref. 43 used a cost-effective scanning acoustic setup, which can simultaneously measure the acoustic signals of commercial Kokam pouch cells. Post-mortem results confirmed that the covering layers (and resulting gas accumulations) observed in the scanning acoustic microscopy were raised from lithium plating. Furthermore, increased covering layer formation on inhomogeneities produced by the cell manufacturer was also observed at high clamping pressures.

LIBs are expected to cycle in environments with changing temperatures, with greater temperature margins in polar regions due to extreme seasonal variations. Thus, to investigate how temperature variations could contribute to aging mechanisms, researchers in ref. 68 cycled the cell at 1C rate (CCCV charge and CC discharge) at a temperature of < 10 °C and then shifted the ambient temperature to different higher temperatures (including 20 °C and 60 °C) at open circuit voltage and 0% SoC. Acoustic signals were recorded during cycling and temperature shifting. Results show that the ToF loss was observed for all temperature shifts due to temperature effects on sound transmission and associated visible gassing. Subsequently, the ToF continued to increase until an abrupt disappearance within one hour of transitioning to 60 °C. This was in contrast to the 20 °C case, where the ToF

**Fig. 2 | Illustration of the requirements for integrating sensors with batteries, highlighting examples of acoustic/ultrasonic sensors and mechanical sensors. a** [Top] The schematic experimental configuration of two acoustic transducers (pulse/listen and listen). The generated sound propagates through various packaging, current collector (CC), electrode, and separator layers of the battery. [Lower] Example illustration of received signals, where generally two peaks are presented in the signal amplitude, and the time of flight (ToF) increases as a function of the state of charge (SoC) during discharge. This shift, represented by the transition from the black line to the red line, results from changes in electrode densities as the SoC ($x$ in the figure) changes. **b** The schematic experimental configuration of three lithium-ion batteries sandwiched between two Garolite end plates. Each battery is $120 \times 85 \times 12.7$ mm with a 5 Ah nominal capacity. The load cell is installed to measure the force from the battery expansion. The end plates are bolted together, while the Garolite middle plate acts as a separator between the cells and the load cell. (Figures **a** and **b** are reproduced with permissions from refs. 33 and 46, respectively).

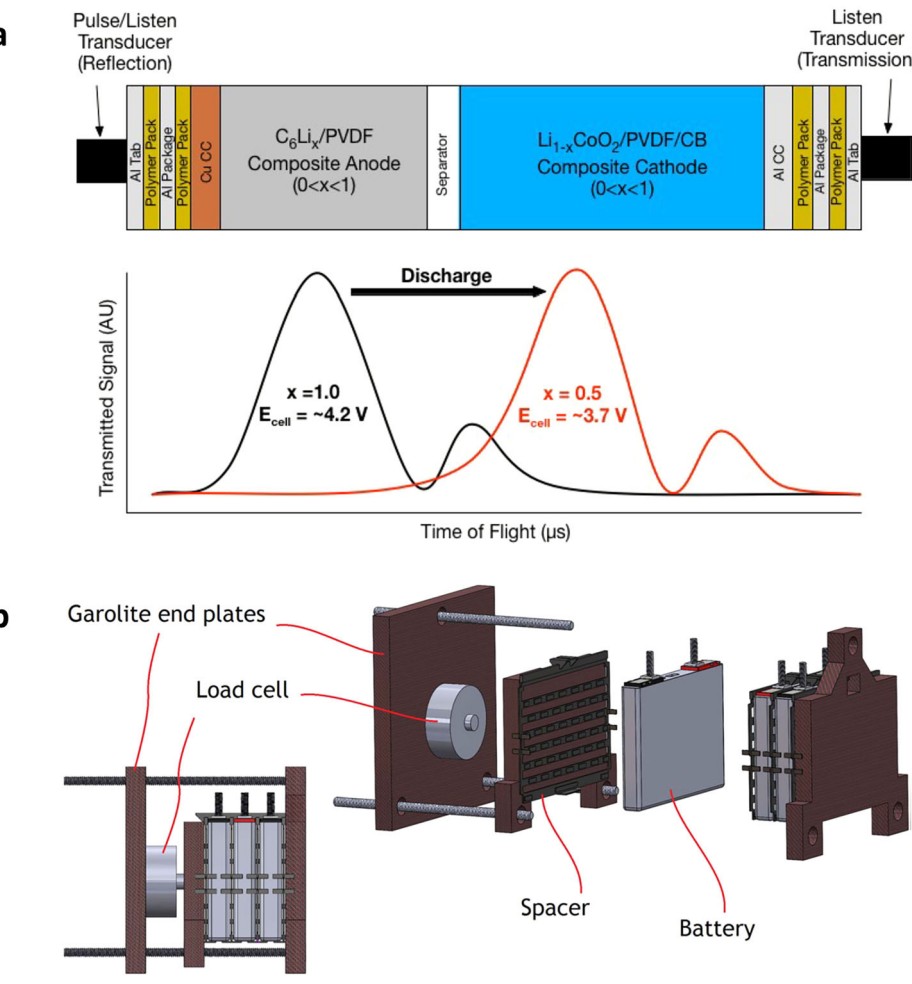

remained relatively constant after the temperature change. The open circuit potential was steady in the shift to 20 °C, whereas it underwent a significant decrease to 0 V in the case of 60 °C, suggesting complete cell failure with signs of gassing. Such attenuation of the signal is believed to come from the lithium plating-induced gassing decomposition reactions, as the cells placed in a 60 °C without cycling or cycled at low rates with a < 10 to 60 °C transition did not experience gassing. Arrhenius's relationship between the rate of acoustic attenuation and the magnitude of the temperature shift was also found, making the signal attenuation time an excellent indicator of when gassing would occur at a given temperature shift. Combined with post-mortem analysis of the anode materials, the author further concluded that the lithium deposition at < 20 °C was correlated with an increase in high binding energy components in the SEI such as lithium carbonate, which disappeared after cycling at 60 °C; and the continued reaction of the Li-rich SEI formed at 0 °C may lead to gassing observed at 60 °C.

Besides studies in lithium plating, research on various abuse scenarios has also been conducted. The experimental results in ref. 69 show that, when the over-charge started, abnormal behavior appeared on the acoustic signal strength, whereas the battery voltage seemed normal. More interestingly, before the acoustic transducer lost contact with the cell surface, acoustic features (including strength and ToF) were not repeatable during continuous over-charge. In contrast, no difference was detected from electro-thermal measurements. Similarly, overcharge tests (up to 5 V) in ref. 35 showed that the ToF and SA could indicate battery failure 0.872 h ahead of battery swelling and 0.817 h earlier than the temperature-based method, providing earlier warning of catastrophic failure and a more extended time margin for failure prevention. Kim et al.[61] detected many burst-type acoustic signals during thermal exposure to 85 °C. In comparison, a tiny number of

signals were detected for the battery exposed to 60 °C, evidence of the microcracking of the cathode material. In ref. 62, high-temperature abuse experiments on NMC cells are performed by measuring the electrochemical impedance spectroscopy (EIS), strain gauges, and ultrasonic waves. Results show that ultrasonic signals are in excellent compliance with the EIS results when the temperature varies from 40 °C to 100 °C and can be potentially used to identify different processes such as SEI dissolution and evaporation of solvents. Similarly, Hao et al.[70] investigated the failure process of cylindrical LIBs under mechanical abuse by monitoring the terminal voltage and acoustic signals. Four stages can be observed during the bending based on the acoustic features, aligning well with force signals, whereas nearly no variation was observed in the voltage.

**Mechanical sensors**

With the need for high energy density and cost-effective LIBs, numerous attempts have been made to include silicon-based materials in conventional graphite anodes. However, the use of Si in commercial anodes is still highly limited due to performance degradation deriving from its intrinsic issues, such as severe volume expansion during cycling and potential stretchable SEI layer during calendaring aging[71,72]. Although the consequences of volumetric changes have largely been overcome using Si particle-size control, the complexity of their SEI layers and physical-based models is still not completely understood[72]. With in-suit atomic force microscopy, Kumar et al.[73] observed extensive cracking during lithiation in the patterned Si island, where a thicker SEI film was formed at the edge and corners than at the center of the island. Furthermore, the reopening and closing of SEI cracks were seen in the first several cycles, suggesting that the surface cracks did not completely close during delithiation, although the underlying Si

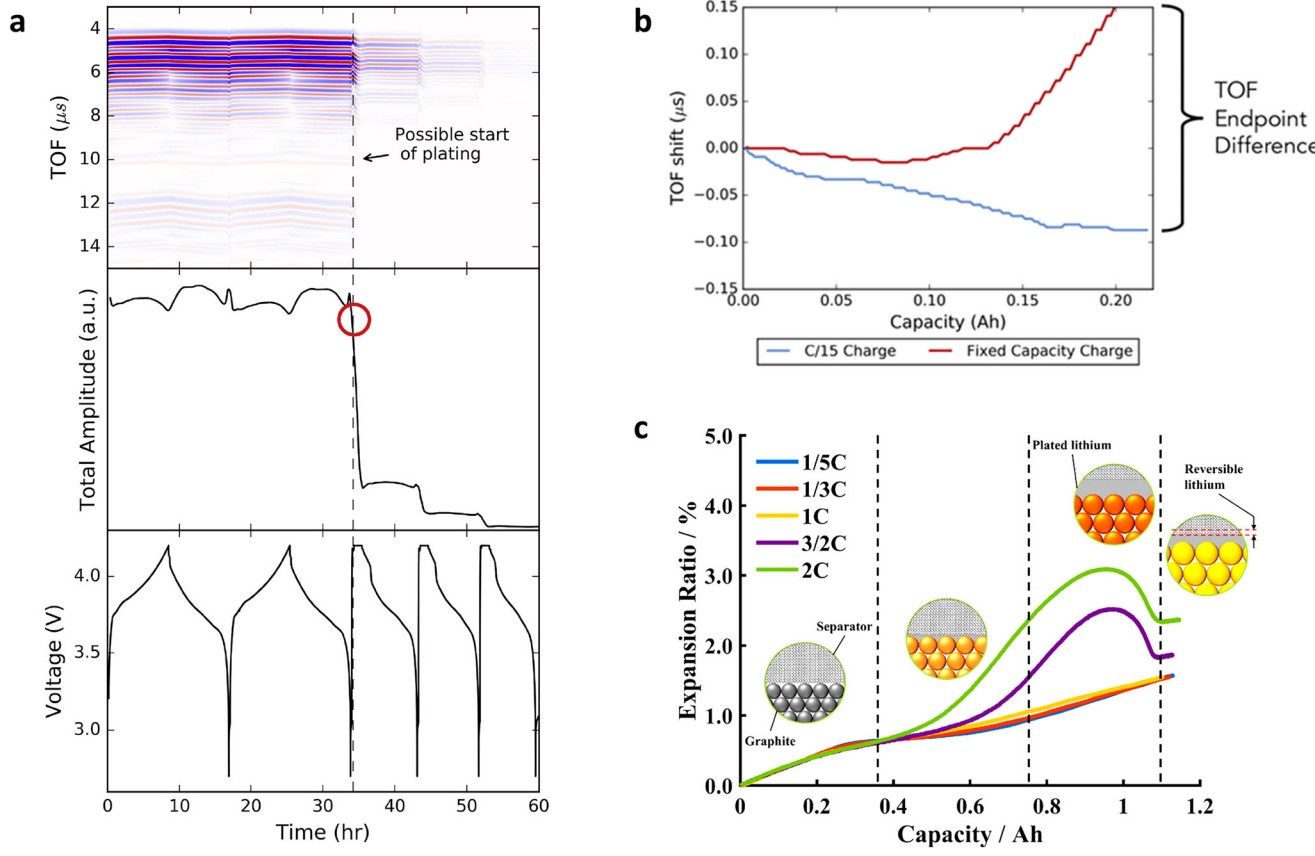

**Fig. 3 | Overview of sensor data and extracted features used for lithium plating detection, with a and b using acoustic sensors and c using mechanical sensors.** **a** Acoustic results of voltage-dependent cycling protocol of a 210-mAh cell at 5 °C, for two cycles at C/10 constant current (CC) charge and discharge, followed by cycles of 1C constant current constant voltage (CCCV) charge and C/10 CC discharge. The acoustic signal quickly attenuates and then disappears upon the first 1C CCCV charge, ascribed to the possible start of Li metal plating. **b** The acoustic time of flight

(TOF) shifts of C/15 CC and fixed-capacity (at 0.210 Ah) charge at 1C charge at 10 °C. When the fixed-capacity charge is finished, the TOF shift is completely different from that of the C/15 charge, with a large difference in the endpoints, indicating potential plating. **c** The evolution of battery volume expansion ratio with the charging capacity at different C-rates. Overshoots in the expansion ratio in higher C-rates suggest the occurrence of lithium plating and stripping. (Figures **a** and **b** are reproduced from ref. 32, and Figure **c** is adopted from ref. 48. All with permissions.)

retracted back laterally. Such high-resolution images can provide important insight into the SEI failure process but are too expensive to be implemented in partial conditions. Instead, fiber optical sensors are a cheaper option to reveal the operando mechanical distributions inside and outside the battery.

For example, in 2016, Bae et al.[30] placed the fiber Bragg grating (FBG) sensors between the anode and separator and directly implanting them into the anode (referred to as 'attached' and 'implanted' respectively by the authors). The studied pouch cells featured a 100 $\mu$m-thick LCO cathode, a 110 $\mu$m-thick anode layer, a 25 $\mu$m-thick separator, and a 100 $\mu$m-diameter FBG. After cell assembling, several charge-discharge cycles were performed, in which the voltage curves did not exhibit significant differences from those of the ordinary cells, indicating that the FBGs did not adversely affect cell performance. The received FBG spectrums revealed two peaks in both attached and implanted cases. More interestingly, repeatable monotonic peak shifting occurred for the attached sensor during charge, whereas more complex shape changes with two peaks occurred for the implanted sensor. This difference occurs because the implanted FBG sensor was entirely surrounded by graphite particles and simultaneously underwent accumulated longitudinal and transverse strains from the expansion and contraction of the anode. In contrast, the attached FBG sensor was subjected only to longitudinal strain or stress, resulting in peak shifting without shape changes. These studies justified that FBG sensors are sensitive to detect the dimensional changes of each electrode. Following a similar idea, the FBG sensor was successfully implemented into commercial battery modules with four cells connected in parallel[56,57]. A semi-empirical model was developed to estimate SoC with the measured strain. However, this pure FBG-based

SoC estimation failed, especially when a flat region (SoC at 60% to 80%) was presented in the wavelength shift v.s. SoC curves. Thus, classical Coulomb-count SoC estimation was included, and the averaged SoC from these two different methods was considered as the final estimated SoC. The combined estimation framework achieved less than 2.5% errors under various temperatures and dynamic cycling profiles.

Though internal mechanical features provide very direct information about internal processes, a larger body of research has used external sensors to detect the mechanical changes (e.g., displacement) of the LIBs. Early in 2015, the authors of ref. 74 applied a fiber optic strain sensor on the surface of a commercial NMC cell. Strain changes during relaxation were seen to increase significantly only at higher SoCs, which authors attributed to the potential inhomogeneous ion distribution in the graphite anode. That is, when charged at high SoC, lithium ions concentrate in the outer electrode region, causing greater volume expansion. After charging, lithium ions are not inserted into the graphite particles but transferred from the outer electrode region to the inner electrode region, resulting in more evenly redistribution in the anode and faded excess volume change. With silicon/graphite NMC pouch cells, authors in ref. 49 measured the cell thickness in a representative C/10 cycle after formation, and it peaked at the beginning of discharge, regardless of the relaxation time between charge and discharge. Similarly to the hypothesis in ref. 74, this expansion peak was explained by the remaining Li⁺ concentration gradients in the anode[37]. However, the galvanostatic intermittent titration technique results revealed that this expansion irregularity likely stemmed from the interplay between anode contraction and cathode expansion.

https://doi.org/10.1038/s44172-025-00383-9                                                                                      **Perspective**

To further investigate and quantify the volume changes, considerable efforts have been devoted to developing a multi-physics model that combines electrochemical, thermal, and mechanical dynamics. In 2016, by splitting the origins of the volume changes into three significant parts: li-ion intercalations, temperature changes, and preload, three different sub-models, namely the equivalent circuit thermal (ECM-thermal), swelling, and force models, were correspondingly built[75]. No preload was considered in the first two models. The intercalation-induced swelling was estimated from the look-up table, where no temperature change was assumed within small current operations. The temperature change-induced force was calculated based on the temperature differences between the core and surface of the cell. The thermal expansions of all components in the setups have also been considered. The final integrated model was validated under no preload and preload conditions, where low errors were obtained in all conditions, confirming the model's validity. Apart from studies at the cell level, the volume changes in porous electrodes will cause either changes in the electrode strain or the electrode porosity[76,77]. Based on the assumption of porous rock mechanics[76], a constant molar volume change due to lithium intercalation was utilized and linked to the electrochemical kinetics through the local volumetric current density in each electrode. The stress, local strain, and overall strain in a porous electrode were then estimated through COMSOL Multiphysics simulation. The porosity distributions, similar to the strain distributions, were seen to be affected by the discharge rate and, in turn, the state of discharge. Later, in 2023, the authors extended the study to calculate the cell thickness considering the multi-species of the cell. The developed model predictions proved to agree with the experimentally measured thickness and captured the primary feature of the volume change profile. Authors in ref. 78 developed an electrochemical thermal-mechanical model, specifically considering the inhomogeneity effect at the electrode level. The coupled mechanical model emphasized lithium concentration gradients as influential stress drivers in electrode particles. Validation results from 0.5C to 5C rates revealed that mechanical effects play a minor role in diffusion and voltage response at low C-rates (up to 2C), while they become more important for higher C-rates. This is because high C-rates cause a significant reaction current density variation in the electrode, leading to lithium concentration gradients in the solid phase and resulting in large stresses at the electrode particle surfaces. In 2023, they extended this study to commercial cylindrical cells with silicon/graphite anode[79], where the volume expansion of the silicon was specially considered. Simulation and experiments reveal that silicon stress remains low at a low depth of discharge but sharply rises beyond 80%. Conversely, graphite stress increases with decreasing temperature and depth of discharge. At higher C-rates, peak stress in the graphite increases as expected. However, this decrease in silicon was caused by earlier hitting of voltage cut-offs, leading to lower active material utilization since silicon is active mainly at high depth of discharge. Such electrode-level-based models can help to simulate the particle cracking-induced volumic changes and estimate the cell thickness variations.

Besides the efforts in using mechanical parameters to develop a multi-physics model, research has also been conducted to track the aging trajectory and potentially interpret the aging mechanisms. With commercial LCO punch cells, cycling tests were performed under different C-rates and temperatures with 0.05 MPa preload in ref. 45, and the collected stack peak stress showed that the stress increased linearly with SoH at 55 °C. However, this linear relationship did not hold for the cells cycled at 22 °C after about 200 cycles, where the SoH decreased without a corresponding peak stress increase. By comparing their DV data, the authors attribute it to an alternative capacity fade mechanism that happened around 200 cycles at 22 °C, while at 55 °C, the $t^{1/2}$ dependence of capacity suggests the SEI growth. An empirical model was also proposed to support the explanation of the linear stress-SoH relationship. Similarly, in 2016, Samad et al.[46] compared the incremental force analysis with the classic incremental capacity analysis for commercial NMC batteries, especially with higher C-rates (e.g., 1C) where the incremental capacity analysis typically does not offer comprehensive insights into battery dynamics[22]. The comparison results show that the

peaks in both incremental curves shift linearly as the number of cycles increases. In 2022, they extended the study by comparing the second differential of expansion with lab-made NMC111 batteries[80]. The cycling aging data under different temperatures and currents show that the differential of expansion features linearly depends on capacity. The shift of these features could be linked to different aging mechanisms. However, no post-mortem analysis was performed to validate the hypotheses further. Schiffer et al.[81] compared the second derivatives of strain over capacity ($d^2\epsilon/dQ^2$) with $dV/dQ$. Experimental results show that electrode phase transitions in $d^2\epsilon/dQ^2$ curves were visible at C/2, 1C, and above rates. Meanwhile, $dV/dQ$ curves only indicated reliable transitions at charge rates less than C/4 for the same cells. By theoretical analysis, the author concluded that the lattice strains/displacements in the electrode were large enough to produce the observed peak at higher C-rates; meanwhile, the peaks in $dQdV$ curves become impossible to measure as the free energy of formations for different phases were compatible smaller. Such lattice strains could accumulate continuously during operation and cause structure degradation and oxygen loss, especially in Mn-rich cells[82]. Consequently, tracking the $d^2\epsilon/dQ^2$ peaks is essential.

The application of mechanical sensors also extends to the detection of lithium plating. As discussed, local plating can happen due to inhomogeneities induced by manufacturing or usage. To detect such a 'bad spot' where plating might happen, Spingler et al.[83] used two laser heads to scan the cell thickness continuously. The Kokam pouch cell and the setup were placed in a controlled thermal chamber. Once the cell was charged with high currents (e.g., 1.5C and 2C), distinct expansion overshoots (amounted up to 0.8% of cell thickness) started shortly before the CV charge and ended before the end of the charge. With the distinct plateaus observed in the voltage relaxation data and post-mortem inspection, the authors attributed such expansion overshoots to irreversible plating. Furthermore, based on the mapping of maximum local irreversible expansion and temperature rise at different C-rates and SoCs, an optimal charging protocol was obtained to minimize the charging time while maintaining minimized irreversible expansion. Compared to the benchmark 1C CCCV charge, the optimized charging protocol reached an 11% reduction in charging time and a 1.6% longer lifetime after 1000 cycles. Similar overshoots due to lithium plating were seen in ref. 48, as shown in Fig. 3c. To further quantify it, a volume expansion model was developed based on the electrochemical model in COMSOL Multiphysics. With the observed nearly constant temperature on the cell surface, the thermal expansion of the studied cell was excluded. The lithium plating and stripping dynamics were considered using the Bulter-Volmer equations, and reaction products were assumed to cover all the anode surfaces, resulting in an adapted porosity. The simulated volume expansion ratio agreed well with the experimental results, and overshoots at high C-rates were clearly presented. Furthermore, the simulated total extra expansion was found to have a good linear relationship with the irreversible expansion and capacity loss, offering opportunities for quantitative detection of plating in practical applications.

Excessive volume changes can occur in various hazardous situations, with thermal runaway being one notable example. Authors in ref. 84 modeled early-stage thermal runaway, which was triggered by an internal short circuit, by considering the thermal dynamics of the core, middle, and surface of the battery. Side reactions, such as electrode and SEI decomposition, were mainly considered and assumed to be the cause of gassing. Additionally, the force changes were estimated by the gas evolution model, which took $CO_2$ as the main component of the vented gas. Experimental results show that a considerable swelling force signal can be detected before the surface temperature rises during the internal short circuit. The developed model can capture quick thermal runaway and slow self-discharge after triggering the internal short circuit with proper parameter tuning. Li et al.[85] employed the internal FBG for temperature and force measurement. They observed that the change in cell force was tens of seconds earlier than the change in cell temperature under nail penetration and thermal abuse tests.

## Gas sensors

LIBs are subjected to different hazards, which potentially result in gas release. Depending on the occurrence of thermal runaway, cell venting can be categorized into first venting and thermal runaway gas venting[86]. The former happens when the internal pressure surpasses a critical value, causing the pressure-burst disk to open. The subsequent thermal runaway venting is more aggressive and releases more vent-gas. Wang et al.[87] used two different cell samples (prismatic and pouch) to compare the venting gas at four different triggering tests, including side-heating, nail penetration, overcharge, and oven heating. Gas chromatography analysis revealed that the main components in the venting gas were $CO$, $CO_2$, $H_2$, $C_2H_4$, and $CH_4$. Overcharging was identified as posing the greatest threat to battery safety among the four tests conducted for both battery types.

Similarly, Cai et al.[36] compared the vent-gas composition under different abuse conditions reported in the literature, including overcharging, overheating, nail penetration, and cell leakage. They found that the major component of the first venting gas was $CO_2$. Furthermore, the $CO$ and $H_2$ were not detected consistently in the first venting events but in the subsequent vent-gas. The volatile organic components, typically produced upon first venting and during cell leakage, were found in large amounts in most cases, highlighting the need for such sensors to detect cell leakage. Based on the significant presence and early occurrence of $CO_2$ in all venting events, the non-dispersive infrared $CO_2$ sensor was selected among four different volatile organic components and $CO_2$ sensors due to its sensitivity and cost-effectiveness. Since venting involves changes in pressure, temperature, and humidity, these sensors were assembled and placed next to the cell inside an unsealed enclosure. The cell was overcharged until the gas venting occurred. Experimental results indicated that the force and voltage experienced immediate changes upon the gas venting. However, a rapid increase in $CO_2$ concentration and a decrease in humidity were observed after venting, while the battery voltage and force remained constant. They also showed that, in a battery pack module, the proposed gas detection system requires only one gas sensor at the outlet of the vent gas channel, further highlighting the effectiveness of these CO2 sensors. Peng et al.[88] investigated the thermal abuse-induced gas hazard in cell-to-pack systems. They used $H_2$ sensors and Fourier transform infrared spectroscopy to analyze the composition of vented gas. Thermocouples and pressure sensors were also used to record gas pressure and temperature. The experimental data indicated that $CO$, $CO_2$, and vented electrolytes were the predominant products. Simulations showed that CO concentration peaked at 8 seconds during the venting process and varied across different locations within the pack. These findings highlight the need for optimal pack design, such as the strategic placement of valves.

The above experimental analysis provides valuable perspectives on the products of gas venting. To further quantify and investigate gas venting propagation, especially during the thermal runaway, Wang et al.[89] proposed a multi-scale multi-phase model. They divided the venting transition process into three stages based on reaction locations. Firstly, as the temperature approaches the onset values, reactions within the cell's jelly roll will be more aggressive, generating more heat and gases, causing the current collector and separator to start melting. Then, with rising temperatures, the organic solvents begin to vaporize, creating pressure accumulation in the headspace. The gas flow entrains the bulk fluid of electrolytes and solid particles, traveling through the headspace to the safety valve. Once the gas momentum is sufficient to open/break the pressure-burst disk, it vents into the ambient, forming a jet flow. Meanwhile, the entrained bulk fluid of electrolyte will fragment and atomize into smaller droplets under the effect of aerodynamic forces. Based on this understanding, three different sub-models were developed. The discrete phase model coupled with the computational fluid dynamics based on the Euler-Lagrange frame was employed to simulate the venting process, where the fluid phase was treated as a continuum while particles were treated as a discrete phase. Furthermore, a lumped thermal model was used to track the thermal runaway temperature variations. The coupled multi-phase model was then validated through thermal abuse experiments by comparing the temperature, jet velocity, and mass loss. The

simulation results further revealed that the electrolyte vapors dominate the gas release before the thermal runaway. To further investigate the venting propagation in a pack, the authors extended their study by including a thermal resistance network to simulate the heat generation of the pack[90]. A similar computational fluid dynamics model was used to address the molecular mass transfer inside the pack. The temperature evolution during thermal runaway propagation and gas concentration after venting agreed well with the experiment measurements. These modeling efforts provide significant insights into the chain reaction and byproducts of the venting process, aiding in the optimal design of battery thermal management systems.

## Discussion

The literature discussed above focuses mainly on the cell level. However, the successful integration of sensors into the vehicle BMS is a critical aspect that requires careful consideration at all levels - cell, pack, and module. Achieving optimal BMS performance requires addressing the challenges associated with sensor assembly. Foremost among these challenges is determining which sensor types to assemble, the exact location of the sensors, and the quantities required at different levels of the battery system.

To tackle this challenge, there are several vital questions one needs to answer first:

1. Does this type of sensor provide distinctive information that cannot be obtained from conventional voltage, current, and single-located temperature sensors?
2. In which scenario, what unique information does this sensor provide?
3. Does the unique information help to better understand, monitor, and control the battery?

As detailed in Table 1, we answered the above questions sensor-by-sensor by providing a convenient organization for papers discussed in this study and others that could not be discussed because of space limitations. From this table, we notice that one of the most significant advantages of using emerging sensors is their ability to elucidate the mechanisms behind various side reactions, particularly those related to safety. Subsequently, their sub-features serve as better indicators for detecting hazardous scenarios compared to those of conventional sensors. For instance, with ultrasonic[69], mechanical[84], and gas sensors[91], the electrical/mechanical abuses can be detected earlier than using conventional voltage or temperature signals, providing additional time for the vehicle BMS to implement active mitigation measures against potential thermal runaway. Moreover, using these advanced sensors also yields higher accuracies in SoC and SoH regression/estimation or multi-physics modeling for LIBs. However, there are still limited quantitative comparison studies on these topics, indicating a need for future research.

We have also noticed that the complexity of the coupled electrical-mechanical-thermal dynamics of LIBs has driven researchers to adopt a holistic approach, which stretches beyond one type of emerging sensor to integrate information across different types of sensors and diagnostic techniques in an interdisciplinary approach. For example, Knehr et al.[31] used the EIS and ultrasonic sensors to study the 'break-in' period between cell formation and steady states. By analyzing the ToF and impedance-fitted parameters, the authors discovered that the full-cell performance can be significantly affected by non-chemical crosstalk between the two electrodes. Specifically, during the post-formation 'break-in' period, side reactions introduce increased swelling of the graphite anode (evidenced by increased ToF) and, accordingly, increased pressure inside the cell. The increased pressure forces the electrolyte to wet previously inactive portions of the LCO cathode, lowering the cell impedance. These findings demonstrate how the interplay between components during early cycles can affect future battery performance. Likewise, Laufen et al.[92] used voltage, strain, and impedance measurements to determine the optimal pressure for enhancing cell cycle lifetime. They found that applying an external mechanical load improved access to smaller pores by pressing in the electrolyte, thereby enlarging the charge transfer area and reducing charge transfer resistance. These

**Table 1 | Table of literature classified by sensor types and objectives**

| | Acoustic & ultrasonic | Mechanical | Gas |
|---|---|---|---|
| General requirements | Close contact to cell surfaces by potentially applying external pressure[43,44], glue[50,51], and water-based gel[52] | Depending on sensor principles and locations, clamping pressure[45–49] and epoxy adhesive[53] might be needed | Cell is placed inside an unsealed enclosure[36] or sealed container/autoclave with a separate atmosphere (e.g., $N_2$)[88] |
| Key parameters tracked in LIBs | Young's modulus[109], mass density and cell thickness[50,110] | Surface strain[53,74], force[46], internal strain[30,55,57] and displacement[64] | Venting gas composition[88], $CO_2$ concentration and humidity[36] |
| Potential features | ToF, SA, signal strength[69], ToF endpoint[32], the first derivative of ToF and SA to the capacity ($dToF/dQ$ and $dSA/dQ$, respectively)[51,66] | Wavelength[30,57,74], stack peak stress[45], incremental capacity to force ($dQ/dF$)[46], differential displacement to capacity[64], differential expansion ratio[48], second differential of strain[81] or expansion[80], and expansion velocity[77] | CO and $H_2$ molar percentage[88], $CO_2$ concentration and humidity[36] |
| Characterization | Spatially[44,65], thickness direction[107], scanning entire cell cross-section[43] | Internal[64], external 1D[74] and 2D scanning[83] | |
| States estimation | SoC[33,50,51,110,111], SoH[35,51,67] | SoC[45,57,112], SoH[45] | |
| Multi-physics based modeling | Theoretical ultrasonic model[52] | ECM-thermal swelling and force model[75], electrochemical mechanical model[48,113], electrochemical-thermal-mechanical model[78,79], internal short circuit model[84] | Gas hazards simulation[88], venting propagation multi-scale multi-phase model at cell level[89] and pack level[90] |
| Degradation mechanisms studies | Irreversible loss of lithium due to SEI layer formation[51,66], electrolyte consumption[114] | SEI growth[45], structural degradation[81] | |
| Nondestructive diagnosis | Lithium plating detection[32,43], venting detection[114], and structure defects detection[44,107] | Lithium plating detection[48,83], thermal swelling[115] | |
| Abuse studies | Thermal abuse[61,62], electrical abuse[35,68,69], mechanical abuse[70] | Internal short circuit triggered thermal runaway[84], mechanical abuse[85] | Electrical abuse[36,87], thermal abuse[87,88], and mechanical abuse[87,91] |
| Distinctive information | Simulated layer-resolved SoC[52], non-repeatable earlier indicator (than voltage signals) for over-charge[69], longer time margin (than temperature-based method) for over-charge induced failure prevention[35], and nondestructive structure defects[44,107] | Electrode lattice strains/displacements[81], earlier indicator (than temperature signals) for internal short circuit triggered thermal runaway[84] and mechanical abuse[85] | Earlier indicator (than voltage signals) for nail penetration triggered venting[91], simulated venting propagation process and byproducts quantitative analysis[90] |

successful proof-of-concept studies demonstrated the efficacy of using a multi-sensor system to diagnose potential side reactions in LIBs. However, validation through post-mortem analysis is essential to correlate sensor data with actual physical changes, thereby confirming the presence and extent of the assumed side reactions.

It is also easy to find that several boxes in Table 1 are rather sparse or entirely empty. Some of these gaps represent opportunities for future research, while others highlight the specific scenarios to which certain types of sensors are applicable. For instance, gas sensors play a crucial role in detecting the presence of gases and analyzing their composition when vented. These sensors are invaluable for identifying potential abnormalities and ensuring safety. However, their effectiveness is limited to scenarios where gas venting actually occurs. In contrast, under most operating conditions, LIBs remain in a safe status, rendering the gas sensors less actively needed. Consequently, while gas sensors are essential for certain safety protocols, their applicability is constrained to specific circumstances where gas emissions are detected. Furthermore, we also see a lot of opportunities to apply these sensor techniques in developing new battery technologies, such as solid-state batteries. Solid-state batteries are often considered safer than LIBs; however, the use of toxic solid electrolytes or the potential leakage of liquid fractions to the anode could lead to additional safety risks, especially thermal runaway[93]. Therefore, whether increased safety exists or not still needs to be more intensively investigated, and the discussed emerging sensors serve as excellent candidates for developing safety test standards for them. Such studies will also help fully exploit the potential of these sensing technologies and establish standardized practices for their integration into commercial battery systems, which we will address in more detail in later sections. Lastly, another notable emerging research avenue is using these sensors to evaluate and sort second-life batteries, as they can assess cell thickness, potential internal cracks and defects, etc., providing significant benefits for battery recycling and repurposing[94].

## Physics-guided methods for battery monitoring and control

First-principle models are used extensively in various engineering and environmental applications. In the BMS domain, based on cell-specific material properties and operational parameters, these models are expected to replicate and possibly predict a selected number of battery characteristics and mechanisms from which effects on cell performance are derived. The equivalent circuit-, electrochemical-, and impedance models are the most used ones, and to improve the model/estimation accuracy, advanced algorithms (e.g., Kalman filters family) could be implemented[95]. Though these methods are based on known physics, in most cases, they require numerous parameters tailored to specific batteries and conditions, which limits their model performance and makes them computationally intensive. When applied to real-world scenarios, such as the cycling-calendaring conditions that vehicle batteries often experience, these models often struggle to remain accurate and require significant refinement to be effective along the aging process. In other ways, ML methods have emerged as powerful tools for battery monitoring and control. They excel at identifying patterns in large datasets, allowing them to predict battery states and performance without the need for explicit physical laws. However, the missing battery principles and large dependency on the quality and quantity of the training datasets could lead to poor model generalizability in new or unexpected conditions, yet unable to provide interpretability, posing further challenges in understanding and trusting their predictions, especially when safety is at stake.

Given these challenges, there is a growing need for hybrid methods, that integrate fundamental physical laws and domain knowledge by 'teaching' ML methods about governing physical rules, which can, in turn, provide strong theoretical constraints and inductive biases on top of the observational ones[96]. The key of these hybrid methods is how the prior knowledge stemming from our observational, empirical, physical, or mathematical understanding of the system can be leveraged to improve the performance of a learning algorithm. To advance this goal, researchers have

identified five key methodologies[97]: a) physics-guided loss function, b) physics-guided initialization, c) physics-guided design of architecture, d) residual modeling, and e) hybrid physics-ML models. When applying these frameworks in the battery field, several critical questions are presented: how to embed physics into ML, and how these physics-guided methods can improve the model's generalizability and interpretability, particularly concerning aging processes. Additionally, it is worth investigating whether emerging sensing techniques can further enhance the accuracy and effectiveness of these methods.

## Recent advances

In 2023, Tu et al.[98] proposed two frameworks, HYBRID-I and HYBRID-II, to integrate the single particle model with thermal dynamics (SPMT) to create a physics-informed feedforward neural network, which is additionally informed of the internal state of the SPMT to be aware of the ongoing dynamics of the physical model and thus learns more effectively what is missed from the physical model. HYBRID-I used neural networks (NNs) to capture the residual error between the SPMT output and the true terminal voltage. At the same time, HYBRID-II relied on NNs to predict the terminal voltage directly based on the SPMT states. The two hybrid models demonstrated high predictive accuracy across a broad range of C-rates as high as 10C, making them promising for various energy storage applications that require high-power charging/discharging.

Similarly to the residual learning used in HYBRID-I method, Kuzhiyil et al.[99] proposed the use of a universal differential equations framework[100] to develop battery models with improved generalizability. The thermal-ECM model with improved diffusion dynamics (TECMD) was used, its voltage prediction was enhanced by adding a data-driven correction term, $V_{NN}$, to capture the residual voltage. The dynamics of $V_{NN}$ were modeled as a new state variable, whose differential equations were governed by NNs with surface SoC, temperature, and current as the inputs. The input choices were based on the domain knowledge that battery voltage dynamics are influenced by the internal lithium concentration and transport mechanisms. Similarly, another NN was used to capture the temperature dependence. Moreover, a two-stage universal differential equation parameterization method was introduced, combining collocation-based pre-training with mini-batch training, to enable the NNs in the proposed model to learn battery dynamics from multiple data sets efficiently. Especially, the Julia programming language was used as it provides strong capabilities in reverse automatic differentiation, which allows the NNs to efficiently compute the gradients of the partial differential equations residuals with respect to the network parameters, enabling effective training of the physics-informed models[96]. Though the developed model achieved better accuracy compared to the standard TECMD model, the authors did not address the potential sensitivity of the universal differential equation parameterization to noise, which could be a concern when using field data rather than controlled laboratory experiments, yet no aging mechanisms have been considered.

To tackle the high nonlinearity and potentially diverse aging patterns observed under accelerated aging conditions, Jia et al.[101] developed a hybrid framework to predict the aging trajectory, especially the knee-point. Therein, sub-features from battery open-circuit voltage, incremental capacity, and DV curves were used to identify potential aging mechanisms (e.g., loss of active material). Then, the physics-guided feature relationship recognition was used to minimize physical inconsistency, such that the knee-point prediction performance can be enhanced with even a small feature matrix. Subsequently, the decision tree regression and classification methods were used for knee-point prediction and diagnosis of the aging mode, respectively. The experiment results suggest that only 14 cells were needed to train the proposed hybrid method and achieved a lifetime prediction error of 2.02% using the first 50 cycles of data, compared to requiring at least 100 cells without using the physical insights. However, similar to other studies relying on features from incremental capacity and DV curves, the proposed framework would be difficult to implement in real-world scenarios, where the acquirement of these curves is restricted.

One challenge in using field data for battery health prediction is the lack of controlled operating conditions that are typically present in laboratory tests, which are essential for consistent and accurate health assessments. While studies, such as those by Aitio et al.[102] and Li et al.[103], have demonstrated that pure ML methods, like Gaussian process regression, can accurately predict the end-of-life of batteries, further research is needed to explore hybrid methods. These hybrid approaches may offer improved robustness and accuracy by combining the strengths of both data-driven and physics-based models.

Apart from these domain-knowledge-based forward models, inverse models are also investigated. The inverse modeling aims to use the possible noise output to infer the intrinsic physical parameters. One example is the inverse design of battery materials with ML[104], where desired target properties of materials are used as input to the model to identify atomic structures that exhibit such properties. Physics-based constraints and stopping conditions based on material properties can be used to guide the optimization process. This approach can lead to faster innovation cycles and more rapid deployment of advanced battery systems.

## Opportunities and challenges with evolving emerging sensor signals

One of the primary motivations for integrating emerging sensor signals into the physics-guided ML is the complementary nature of their strengths. Prior research has demonstrated that physics-guided ML methods offer opportunities for improved accuracy, robustness, and generalizability. Meanwhile, we have seen from previous discussions that each type of emerging sensor can provide unique physical information about battery dynamics, which conventional sensors fail to do. Thus, by integrating these two approaches, we can leverage the generalizability of physics-guided ML while reducing reliance on assumptions and approximations of the complex dynamics inside the batteries. For example, with external mechanical sensors, previous studies[48] simulated the plating-induced cell expansion by assuming a constant temperature distribution and a uniform plated lithium on the electrode surface, which may not apply in most battery operations and lacks model uncertainty quantifications. The physics-guided ML methods, for example, surrogate modeling[97], can incorporate ML to capture the nuances that the physical models might miss. This reduces the need for assumptions by allowing the surrogate to adapt based on data, rather than being strictly defined by theoretical simplifications. Furthermore, techniques such as the Monte Carlo method allow the surrogate model to provide uncertainty estimates alongside its predictions, reflecting the confidence level in the model's approximations. Another opportunity lies in providing new frameworks for non-destructive aging mechanism identifications, especially by feeding the sub-features from these signals as new inputs to an ML method and using them to infer the unobserved internal properties and states. For instance, acoustic and ultrasonic sensing techniques have shown great potential for monitoring structural changes within batteries. These sensors can detect subtle mechanical changes and the formation of gas bubbles, which are critical indicators of internal degradation processes. When combined with ML, these acoustic signals (time series, 2D or 3D images) can be analyzed as part of an inverse modeling approach, similar to computer vision problems where the goal is to reconstruct and interpret the internal state of the battery from localized internal or external measurements. This approach allows for the early detection of issues such as lithium plating or dendrite formation, which can lead to capacity loss or even safety hazards if not addressed.

Another promising aspect of this integration is motivated by the ability of these sensors to provide faster and earlier detection of potential safety issues during abuse testing. Unlike standard aging tests, experimental abuse testing is more time-consuming and costly, limiting the availability of data for developing and validating physics-based models. Furthermore, the inherent complexity of battery systems makes it challenging to model and understand the propagation processes necessary for diagnosing and preventing such issues. To address this, physics-guided ML simulations can be employed to augment real-world data, generating synthetic data points that

adhere to physical laws. These more accurate models can also enhance risk assessment and provide better decision-making support for safety issues, moving beyond the limitations of purely empirical evidence. In summary, we envision that merging principles from physics-guided ML and new battery measurements will play an invaluable role in the future of the battery field, especially in addressing the pressing safety and performance challenges.

Though potential advantages are seen, challenges are presented as well. One of the primary challenges lies in ensuring the quality and quantity of data available for training and validation. As previously discussed, data from emerging sensors is often collected under controlled laboratory conditions, involving well-designed setups and simplified operations, such as constant current charging and discharging cycles. While these conditions are ideal for initial testing, they limit the variability and complexity of the data, making it difficult to capture the wide range of real-world operating conditions. This can lead to potential biases in the models, and the resource-intensive nature of gathering sufficient high-quality data under controlled settings poses a significant barrier to scalability and practical deployment. The challenge extends further when attempting to generalize models trained on laboratory data to real-world environments. Real-world battery usage is characterized by variability in operating conditions, noise, and unforeseen events that are not typically present in controlled lab setups. As a result, models may overfit the lab data and underperform in practical applications.

Another significant challenge is the selection and engineering of the most relevant features from the wealth of data provided by emerging sensors. Not all signals or features contribute equally to model accuracy, and the inclusion of irrelevant or redundant features can degrade performance, increase computational complexity, and reduce interpretability. Furthermore, combining features from different sensor modalities introduces additional complexities, such as differing noise levels, sampling rates, and signal sensitivities, which require sophisticated preprocessing and alignment techniques. The integration of data from multiple sensors also presents challenges, as sensor fusion techniques are essential to create a coherent and unified dataset. This process involves synchronizing data streams, handling different data formats, and managing varying levels of data quality. Poorly executed sensor fusion can introduce noise and distortions, negatively impacting ML model performance.

In conclusion, while combining physics-guided machine learning with smart sensors offers considerable potential, it also presents significant challenges. One crucial aspect of advancing this field is encouraging the sharing of sensor data and code. This openness is essential for minimizing several risks, such as the lack of reproducibility, overfitting models to limited data sets, and poor generalization to real-world scenarios. Without shared datasets and standardized code, there is also a risk of isolated development, particularly sensor development, which we will discuss in the next section. By promoting transparency and collaboration, the accuracy, robustness, and scalability of these systems can be significantly improved, ensuring future advancements benefit from collective insights and validation.

## Incorporating smart sensing into future BMS

Building on the challenges and opportunities previously discussed, we see that smart sensors have the potential to significantly enhance real-time monitoring and management during hazard scenarios by providing precise and actionable data. Therefore, integrating smart sensing technologies into future BMS is crucial. However, moving from the conceptual stage to the practical deployment of these sensors within BMS requires navigating multiple stages, each with its own set of challenges. As Frith et al.[3] highlight, the technology readiness level (TRL) framework in the battery domain outlines this progression. The process begins with basic research (TRLs 1-2), advances to proof-of-concept and initial prototyping (TRLs 3-4), and ultimately leads to scaled-up manufacturing and vehicle-grade performance (TRLs 6-8), with widespread adoption occurring at TRL 10. This section will follow the TRL scale, discussing successful case studies of smart sensing systems developed at lower TRLs (1-4) and exploring chances and strategies for upscaling these technologies for real-world applications.

### Sensor development and assembling at early TRL stages

During these stages (TRLs 1-4), the focus shifts from conceptualizing smart sensors to developing and assembling them under controlled conditions to ensure they meet the necessary standards for future scalability. Notable examples, as illustrated in Fig. 4, have explored the technical viability of integrating several different types of sensors. In the first case study (as indicated in Fig. 4a), the spatially distributed mechanical, ultrasonic, temperature, and operando odd random phase EIS sensors are attached to the cell surfaces and tabs. A customized printed circuit board is designed to collect and process the sensor data, and further deliver the data to the cell management system. In contrast, the second case study injected sensor layers into the NMC cells (as shown in Fig. 4b). The target multi-sensor platform aims to measure in real-time the internal battery cell temperature, pressure (e.g. mechanical strain, gas evolution) conductivity, and impedance (separately for the anode, cathode, and electrolyte) with customized BMS[105]. Based on the public document from the project, the printed reference electrodes based on lithium titanate or lithium iron phosphate were fabricated and integrated into 3–5 cm² pouch cells successfully. Electrochemical characterization showed the reference electrodes did not significantly impact the pouch cell performance and remained stable for over 400 hours, offering chances for impedance and potential measurement of individual electrodes. However, the author did not specify how the other sensors (e.g., the internal temperature and pressure sensors) function after electrolyte injection. In the last case study[106], to enhance the energy density and cost-effectiveness of EVs and other electrical systems, the authors developed Multifunctional Energy Storage Composites, featuring innovative structural batteries integrated with in-situ networks of sensors and actuators (as illustrated in Fig. 4c). A vertical material integration technique was introduced to minimize sliding between the internal layers of the battery. Unlike conventional designs, where anode and cathode layers are stacked alternately, this new approach encapsulates LIB materials within structural carbon-fiber-reinforced polymer "facesheets". Through-thickness polymer reinforcement pins are used to interlock the layers, enabling efficient load transfer mechanically. Additionally, the sensor networks incorporate distributed temperature sensors, strain gauges, and PZT ultrasonic transducers in a pitch-catch configuration. The goal is to make these components work together to gather supplementary data for real-time estimation of SoC and SoH. Laboratory experiments demonstrated that guided-wave-based ultrasound techniques significantly improved the accuracy of SoC estimation and SoH identification. We notice that although a multi-sensor system was the goal of these studies, none of the cases, as of this writing, have demonstrated the validation of all sensors and components functioning together. Instead, most reports show that each sensor type was tested individually, despite the aim for simultaneous operation. This lack of integrated validation is due to the complexities of smart sensor network integration, both internal and external. Key challenges include stability, reliability, and the lifetime of the sensors themselves, the need for precise sensor placement and insulation, potential reductions in cell energy density, and safety risks, etc.

When integrating sensors into battery cells, the sensor materials must meet strict requirements for chemical stability, compatibility with active materials, and mechanical robustness to ensure reliable performance without compromising safety or cell integrity[105]. Materials prone to corrosion by electrolyte components, such as hydrofluoric acid from electrolyte decomposition, should be avoided. For example, silica fibers used in FBG and luminescence sensors can be corroded by hydrofluoric acid. Additionally, sensor materials must not react with highly reactive substances like lithium metal or silicon. Proper electrical insulation or a secondary separator is recommended to prevent unwanted electrochemical reactions. Lastly, sensor materials should accommodate the mechanical stresses caused by the volume changes of the anode during cycling to prevent high-impedance short circuits. Once the sensor materials are selected and fabricated, the injection and assembling of sensors can introduce potential weak spots that could compromise the cell performance and safety. As highlighted by ref. 105, the sensor tips and measuring leads inserted between the anode and

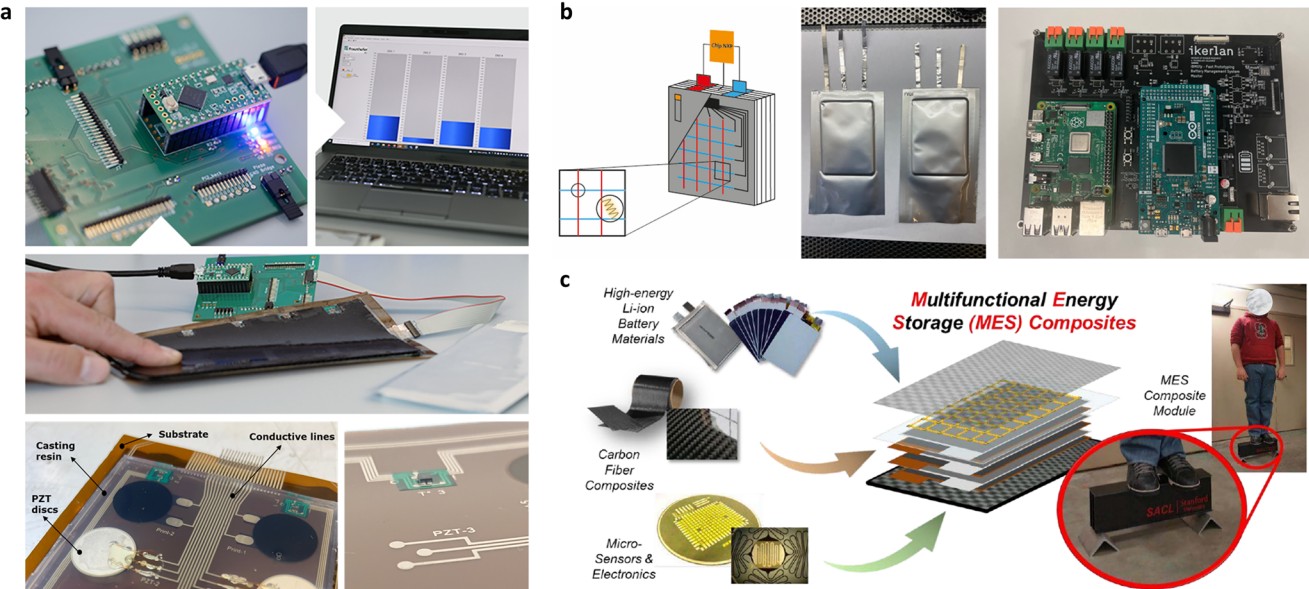

**Fig. 4 | Achievements at early technology readiness level (TRL) stages of implementing a multi-sensor system in lithium-ion batteries (LIBs). a** The spatially resolved external multi-sensor system for commercial pouch cells in the SPARTACUS project, including the printed circuit board for sensor data transforming [Top left], data visualization and storage [Top right], the spatially mechanical sensor layer [Middle], the ultrasonic transducer layer (PZT stands for lead zirconium titanate ceramics) [Lower left], and mounted temperature sensor [Lower right] (Modified based on ref. 116, with permission, Copyright for Fraunhofer Institute for Silicate Research). **b** The schematic distribution of injected sensors (including temperature, pressure sensors, and reference electrode) [Left] and its successful prototyping of 1 Ah cells (with size of approximately 3 cm²) before electrolyte filling [Middle] with the developed battery management system (BMS) Master [Right] in the SENSIBAT project (Adopted from ref. 105, with permission). **c** Multifunctional Energy Storage Composites concept - embedding LIBs materials inside high-strength carbon-fiber composites, together with in-situ networks of sensors and actuators (Adopted from ref. 106, open-accessed).

cathode can increase the probability of high-impedance short circuits due to the expected expansion of the anode. This can result in a high self-discharge rate and low coulombic efficiency of the cell. Additionally, the lack of gas and moisture tightness of the feedthroughs for the multi-sensor platform can lead to premature cell and sensor aging, as well as potential failure of the cell and sensor. Lastly, the sensor injection can further complicate the already challenging cell manufacturing process, especially the risks of defects generated during the rolling and gassing steps.

In contrast to the challenges of injecting internal sensors, attaching external sensors to cell surfaces introduces different complexities, especially in managing sensor placement, size, and operations. Ultrasonic transducers, for instance, are available in various configurations, most of which share a common structure known as a transducer stack, where piezoelectric materials are paired with matching layers and backing materials for mechanical damping. Additional components include wiring, electrical impedance matching, and a protective casing[60]. A signal generator and power amplifier are needed to generate and send the ultrasonic signal to the discs or probe. To ensure the received signal is large enough to be recorded but does not exceed the transducer's maximum amplitude, the transducers must be adjusted for the appropriate thickness range[107]. Furthermore, as the technique reveals structural changes only in the area between transducers, precise sensor placement is critical. Especially for LIBs, which vary in shape depending on the manufacturer, multiple transducers or scanning acoustic microscopy can be used to study potential spatial heterogeneities[32,43]. Additionally, securely fastening the sensors to the cell surface is essential. This can be achieved using methods such as adhesive application, as outlined in Table 1. These requirements highlight the importance of choosing the proper sensor location while also indicating the limited flexibility of these sensors. Moreover, when using adhesives or external pressure, one must account for potential volume changes during battery aging. The attached sensors must be able to adapt to these mechanical variations to ensure long-term reliability. The same considerations apply to mechanical sensors, which also require secure attachment for effective and reliable operation, as discussed in Table 1.

In summary, the case studies highlight promising advancements in developing a multi-sensor system prototype, aiming to improve battery safety. However, as these prototypes were primarily designed for proof-of-concept purposes, issues such as over-engineering, lack of transparency, and limited reproducibility present significant risks of over-extrapolation. To scale these technologies for large-volume manufacturing, further adaptation is required, particularly in optimizing materials, sensor integration, and production processes. Incorporating more established sensor technologies, such as impedance sensors, which have proven effective in thermal runaway detection[108], could enhance system reliability. Broader validation across varied conditions is also necessary to ensure real-world applicability and scalability.

## Challenges in upscaling of multi-sensor system into vehicle BMS

A rapid transition to EVs will require bringing to market more affordable models[1]. LIBs have seen significant improvements in performance and cost, becoming the go-to solution for many applications. Meeting the growing demand for advanced battery management solutions in a variety of applications requires a balance between performance and cost-effectiveness. Consequently, the cost of assembling extra sensors remains important and is expected to be less. For example, in 2017, Raghavan et al.[56] assembled FBG into commercial LG battery modules, where minor machining was done on the cell retaining frames to accommodate the exit of the fiber optical sensor cable. The cost assessment reveals that the cost of the fiber optic cable with multiplexed FBG sensors was projected to drop to < 5 US dollars per meter for mass volume production. However, the overall system cost was still estimated between 100 and 500 US dollars, depending on the volume of production. Furthermore, as the number and type of sensors increase, so do the expenses related to hardware and software system developments, as well as long-term maintenance. This highlights the challenge of reducing sensor costs to a point where large-scale deployment becomes economically feasible.

In addition to cost, the question of sensor assembly must be addressed. Deciding who is responsible for assembling these sensors and where the process should take place is not straightforward. Sensor integration requires

specialized technical knowledge, not just in sensor technology but also in the specific demands of battery systems. Manufacturers might choose to handle the sensor integration in-house during battery production, ensuring tight control over quality but potentially increasing production costs. Alternatively, third-party suppliers could manage sensor integration, but this would introduce logistical challenges, such as ensuring seamless coordination between sensor manufacturers and battery producers. Furthermore, there's the question of whether assembly should occur at the battery cell or pack level, as each approach has different implications for complexity and cost efficiency.

One promising avenue to address these challenges lies in battery swapping stations, which are increasingly being used in some EV markets. These stations offer a unique opportunity for implementing an independent multi-sensor system. When battery packs are periodically removed for recharging or replacement, technicians can conduct thorough safety checks using the independent multi-sensor system. This is analogous to a yearly health check-up, where critical diagnostics are performed to assess the condition of the battery and detect any signs of degradation or potential failure. By leveraging these stations, the multi-sensor system does not necessarily need to be assembled to the EV and continuously work during vehicle operation, simplifying the process while still delivering vital health monitoring.

Beyond swapping stations, there are additional opportunities to apply multi-sensor systems. For instance, embedding sensors directly into battery packs for in-service diagnostics could provide real-time monitoring during vehicle operation, helping to identify issues before they escalate. This could also play a key role in recycling or repurposing batteries, particularly with the concept of 'battery passports,' where sensors can assess the remaining capacity and health of used cells, ensuring efficient second-life applications, such as grid storage. Additionally, in fleet management, multi-sensor systems could enable predictive maintenance, reducing unexpected downtime and optimizing vehicle availability. In the case of post-crash evaluations, sensor data could help determine the structural integrity of battery packs, offering insights into whether repair or replacement is necessary after an impact.

In conclusion, the upscaling of the multi-sensor system into future vehicle BMS presents substantial opportunities alongside significant challenges. More studies in low TRL prototyping are needed to further explore and demonstrate the benefit of combining different sensor types. Attentions are also needed to overcome the high cost of sensors, complexities in assembly, and effective integration strategies. Additionally, conducting thorough customer analysis is vital to ensure that the systems meet market needs and expectations. By addressing these challenges, exploring innovative applications like battery passports, and optimizing sensor integration, the industry can advance toward realizing the full potential of multi-sensor systems in vehicle BMS, ultimately advancing battery technology and safety.

## Conclusions

This study offers a comprehensive examination of emerging sensor applications in battery modeling and monitoring. We have reviewed the fundamental requirements, potential measurements, and key battery features that these sensors can track, providing a thorough summary of their capabilities and unique attributes. Additionally, we explored recent advancements in physics-guided machine learning within the battery domain, highlighting both the opportunities and challenges associated with integrating these new sensor techniques. The aim is to develop more robust models and enhance the detection of potential safety issues. The idea of a multi-sensor smart system is discussed with a detailed analysis of three case studies, which examine both internal and external sensor integrations. These proof-of-concept studies have provided valuable insights and practical experiences that will inform future research and developments in BMS. We would also like to highlight the critical need for sharing sensor data and code, as well as continuous innovation and collaboration, as the dynamic nature of energy storage technologies requires battery modeling and monitoring methods to be adaptive and forward-looking. By addressing the outlined challenges and

leveraging the opportunities presented, the industry can move towards more effective and reliable battery management solutions.

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

## Acknowledgements

The authors would like to acknowledge funding support from the European Union's Horizon 2020 research and innovation program under Grant Agreements No 957221 and No 101103702.

## Author contributions

X.Z.: conceptualization, writing - original draft, reviewing, and editing. M.B.: conceptualization, writing - reviewing and editing, funding acquisition, and supervision.

## Competing interests

The authors declare no competing interests.
