## [Transparent Peer Review file · Communications Engineering]

Emerging sensor technologies and physics-guided methods for monitoring automotive lithium-based batteries

Corresponding Author: Professor Maitane Berecibar

Version 0:

Reviewer comments:

Reviewer #1

(Remarks to the Author)

This manuscript provides a comprehensive insights into the application of emerging sensor technologies and physics-guided methods for automotive battery management systems. This work is well-written with desirable structure. Some suggestions and advices are as follows:

- (1) It is recommended to check the word count of the abstract, which should not exceed the journal's limitation.
- (2) There are some grammar errors and typos throughout the manuscript. Please carefully read the manuscript to avoid these issues.
- (3) Please give the full name of the abbreviation when it first appears, such as: IP in page 5.
- (4) There should be a space between number and unit, such as: 55°C.
- (5) Please remove the unit . in the title of Section 4.3.1.
- (6) In Section 4, it seems like all the content is related to emerging sensor technologies, and the information of physics-guided methods is lacked.

Reviewer #2

(Remarks to the Author)

The authors present a review of vehicle battery state monitoring as well as a future perspective on both sensor technologies and physics-guided methods. The subject of the review is timely and appropriate for a review and perspective article in the field.

It is recommended that the authors make improvements to strengthen the contribution. Some areas for potential improvement to consider include:

- 1) The section discussing gas release monitoring was particularly short and without full details or references. Expansion of this section is recommended.
- 2) The authors do not address the question of sensor costs in any quantitative sense, despite the critical importance for practical battery system monitoring. Cost requirements and constraints should be discussed for the overall monitoring system.
- 3) The discussion of sensor technologies is general and lacking specificity. Some detailed examples of potential sensor types and classes would strengthen the discussion.
- 4) Figures are quite high level and schematic, lacking technical specificity. Additional technical depth would strengthen the overall quality of the manuscript.

Reviewer #3

(Remarks to the Author)

The manuscript presents a comprehensive review of using emerging sensor technologies and physics-guided methods for vehicle battery state monitoring. The manuscript reviewed previous efforts in developing state monitoring technologies and emphasizes smart sensors as a future development direction. However, a few problems need to be addressed before the manuscript is published.

1. The author emphasized in the Introduction that "As a result, these battery models are often limited to specific operating

ranges." However, the authors did not explain in the paper how emerging sensors based state estimation and fault diagnosis technologies address the "out of distribution" problem.

2. Some parts of the manuscript contain well-known information from the literature, which could be reduced to focus more on critical analysis and new insights.

3. State monitoring techniques based on emerging sensors share similar workflows with traditional sensors, including threshold selection, feature extraction, parameter identification, and building regression models. It is suggested that the authors make a more detailed comparison between these two technologies and explain why state monitoring based on advanced sensors is poised to become a crucial avenue for future developments.

4. The value of optical fiber sensors includes simultaneously monitoring the mechanical performance and temperature distribution of lithium-ion batteries. The installation of optical fiber sensors (OFS) is a crucial aspect of fiber-based state monitoring technologies. I suggest that the authors provide a more detailed discussion of these aspects.

5. I suggest that the authors supplement the content related to temperature sensors in Chapter Four. Additionally, Electrochemical Impedance Spectroscopy (EIS) is a mature electrochemical analysis method with commercial implementation solutions suitable for BMS, such as commercial chips and Control Units. EIS can also capture valuable electrochemical information within the battery. Therefore, it is recommended that the authors include additional content related to EIS.

6. It is recommended that the authors supplement the working principles of these sensors monitoring the internal state of the battery in Chapter Four. For example, elucidate how the internal information within the battery influences the signal variations detected by the sensors and how this information reflects battery degradation.

Version 1:

Reviewer comments:

Reviewer #1

(Remarks to the Author)

The authors have revised the manuscript based on my comments. I don't have further suggestions.

Reviewer #3

(Remarks to the Author)

this paper can be considered for publication

Reply to **Reviewers** on

**Emerging sensor technologies and
physics-guided methods for monitoring
automotive lithium-based batteries**

Manuscript number: COMMSENG-24-0053

Xia Zeng, Maitane Berecibar

September 2024

We sincerely acknowledge the insightful and comprehensive comments from the editorial team and all reviewers, which have significantly helped us to improve the quality of the manuscript. We deeply apologize for the delay in submitting our revised manuscript. We have invested considerable effort into addressing the concerns and suggestions raised, and we are very grateful for your patience during this process. In response to reviewers' comments (highlighted in blue), we have made the following revisions (detailed in black). We have also thoroughly proofread the entire manuscript to correct typographical or grammatical errors.

Before answering each reviewer question-by-question, we would like first to summarize the changes in the outline of the manuscript, as these were concerns in inadequate prospective discussions, insufficient information of physics-guided methods, lack of technical specificities in figures, and not strong enough shreds of evidence for proposing emerging sensors as a promising crucial avenue for future developments, etc. Considering all these valuable feedbacks, we have revised the outline of the manuscript as follows:

1. Previous Section 2 - *Overview in battery states monitoring with conventional sensors*, which discussed the physics-free, physics-based, and hybrid methods, has been removed. The physics-free and physics-based methods now are only briefly mentioned in Section 1 - *Introduction*. The hybrid methods are now included in a new section - Section 3, *Physics-guided methods for battery monitoring and control*.
2. Previous Section 3 - *Emerging sensing techniques in battery monitoring* is now Section 2. As pointed out by one reviewer, the focus of the literature

review for each type of sensor has been switched on how they can help in addressing the 'out of distribution' problems, which conventional sensors failed to do. The content on the gas sensors has also been enriched, as suggested.

Additionally, a new sub-section - *2.5 Discussion* has been added to strengthen the technical specificity. Therein, a table (referred to as *Table 1* in the manuscript) is provided to compare and summarize the requirements, application cases, and unique advantages of the discussed state-of-the-art acoustic, ultrasonic, mechanical, and gas sensors. On top of it, we examined the potential benefits of using a multi-sensor system, as well as future research opportunities based on the rather sparse or entirely empty boxes in *Table 1*.

3. A new Section - *Physics-guided methods for battery monitoring and control* is now included as Section 3. As suggested by the reviewer, the recent advances of physics-guided machine learning applications in battery modeling, as well as opportunities and challenges of combining both emerging technologies and physics-guided methods, are discussed here.

4. Previous Section 4 - *Incorporating smart sensing into future BMS* has been refined with the requirements and challenges of moving from the conceptual stage to the practical deployment of these sensors within BMS. Using the Technology Readiness Level (TRL) framework, we examine three proof-of-concept case studies that lay the groundwork for prototyping a multi-sensor system. The practical requirements and experiences in sensor development at low TRL levels are summarized.

Then, the discussion extends to the challenges of upscaling the multi-sensor system for future BMS. As suggested by the reviewer, the balance between performance and cost-effectiveness is highlighted with a cost-analysis study on Fibre Bragg Grating (FBG) sensors. Opportunities for incorporating more mutual sensor types, for example, impedance sensors, are also discussed, as suggested by one reviewer.

In summary, in the current manuscript, the technical sections (2, 3, and 4) started with a literature review, recent advances, and case studies to build the ground of each section, then specific discussions are followed to provide the prospective aspects (challenges and opportunities) for using a single sensor, a multi-sensor system, and incorporating it/them into physics-guided methods, as well as into future BMS.

We would also like to mention that references are used in responses, whose numbers are not the same as in the manuscript itself. Additionally, section numbers have been included in the manuscript to facilitate understanding and reference. However, we note that the author's guidelines recommend avoiding the use of section numbers. We are prepared to remove them at a later stage of the review process if necessary.

Reply to Reviewer 1

General remarks) This manuscript provides a comprehensive insights into the application of emerging sensor technologies and physics-guided methods for automotive battery management systems. This work is well-written with desirable structure. Some suggestions and advices are as follows:

We sincerely thank you for evaluating the manuscript and the positive comments on our work.

Remark (1) It is recommended to check the word count of the abstract, which should not exceed the journal's limitation.

Thanks for pointing this out. The abstract has been revised and is now within the limit of 200 words, as the submission system suggested.

Remark (2) There are some grammar errors and typos throughout the manuscript. Please carefully read the manuscript to avoid these issues.

Sorry for the inconvenience it might cause. For the current manuscript, we have thoroughly proofread it to correct any typographical or grammatical errors.

Remark (3) Please give the full name of the abbreviation when it first appears, such as: IP in page 5.

We appreciate your suggestions on it. The abbreviations used in the current manuscript have been double-checked. The list of abbreviations is not yet included in the current manuscript due to the page limitations but could be provided during the review process if necessary.

Remark (4) There should be a space between number and unit, such as: 55°.

Thanks for the hints. The units used in the manuscript have been double-checked to make sure a space is used between the number and unit.

Remark (5) Please remove the unit . in the title of Section 4.3.1.

Thank you for pointing it out. The title of each section and sub-section has been updated and double-checked not to include the unit.

Remark (6) In Section 4, it seems like all the content is related to emerging sensor technologies, and the information of physics-guided methods is lacked.

Thanks for your insightful suggestions. In the current manuscript, Section 4 mainly focused on discussing the practical requirements, difficulties, and potential solutions of incorporating smart sensing into future BMS. Section 3 discusses the recent advances in physics-guided methods. So, we found it might fit better to include the link between emerging sensor technologies and physics-guided methods in Section 3. Accordingly, a new sub-section 3.2 - *Opportunities and challenges with evolving emerging sensor signals* has been added.

Reply to Reviewer 2

General remarks) The authors present a review of vehicle battery state monitoring as well as a future perspective on both sensor technologies and physics-guided methods. The subject of the review is timely and appropriate for a review and perspective article in the field. It is recommended that the authors make improvements to strengthen the contribution. Some areas for potential improvement to consider include:

We sincerely thank you for going through our manuscript and the invaluable insights you offered.

Remark (1) The section discussing gas release monitoring was particularly short and without full details or references. Expansion of this section is recommended.

We fully agree that gas release monitoring is an interesting research topic to dive deeper into. As recommended, we have enriched Section 2.4 - *Gas sensors* by including more references studying the composition of vented gas from different triggers (side-heating, nail penetration, overcharge, and oven heating, etc) [1,2] with the Non-Dispersive Infrared (NDIR) CO₂ sensor [2] and H₂ sensor [3] in single cells and cell-to-pack systems.

Furthermore, modeling efforts from the literature are also discussed to understand and quantify the venting propagation process [4, 5] by combining the lumped thermal model and the discrete phase model coupled with the computational fluid dynamics (CFD). The coupled multi-phase model was then validated through thermal abuse experiments by comparing the temperature, jet velocity, and mass loss. The simulation results further revealed that the electrolyte vapors dominate the gas release before the thermal runaway. These modeling efforts provide significant insights into the chain reaction and byproducts of the venting process, aiding in the optimal design of battery thermal management systems.

Remark (2) The authors do not address the question of sensor costs in any quantitative sense, despite the critical importance for practical battery system monitoring. Cost requirements and constraints should be discussed for the overall monitoring system.

Thank you for raising this important point. We fully agree that achieving a balance between performance and cost-effectiveness is crucial for the broader

adoption of emerging sensors in future applications.

In sub-section 4.2 - *Challenges in Upscaling Multi-Sensor Systems into Vehicle BMS*, we have emphasized the significance of cost assessments when considering the integration of additional sensors. To provide a clearer understanding, we have included the analysis by Raghavan et al. [6], which estimates that the cost of fiber optic cables with multiplexed Fibre Bragg Grating (FBG) sensors could drop to below 5 US dollars per meter with mass production. However, the overall system cost remains between 100 and 500 US dollars, depending on production volume. This analysis represents, to the best of our knowledge, one of the few cost assessments that addresses the scaling of additional sensors for high-volume production.

Remark (3) The discussion of sensor technologies is general and lacking specificity. Some detailed examples of potential sensor types and classes would strengthen the discussion.

We sincerely thank you for raising this valuable point.

In response, we have significantly expanded the discussion in sub-sections 2.2, 2.3, and 2.4 by enriching the literature review of each sensor with detailed examples of their applications. These include state estimation, multi-physics modeling, degradation mechanism studies, in-situ diagnosis, abuse studies, etc. Specifically, we have highlighted the necessary preconditions for sensor use, the underlying sensor principles, the potential features derived from sensor signals, and how these features correlate with key battery dynamics and phenomena.

For instance, detecting lithium plating in real-time remains challenging with conventional methods. When irreversible lithium plating occurs, either locally or globally, it can cause changes in the battery material density, stiffness, cell thickness, etc. These changes, in turn, affect the propagation speed of sound waves [7] and induce expansion overshoots [8]. Acoustic, ultrasonic, and mechanical sensors can capture these effects through distinct features, such as endpoint differences [7] and expansion ratios [8]. To make these discussions more illustrative, we have included a new figure (referred as to *Figure 1* in the manuscript) that explains the mechanisms of lithium plating, experimental configurations for the two sensor types, and examples of how they can detect plating.

Furthermore, for clarity and better understanding, *Table 1* on page 15 has summarized the discussed literature, classified by sensor types and objectives. From this table, we notice that one of the most significant advantages of using emerging sensors is their ability to elucidate the mechanisms behind various side reactions, particularly those related to safety. Especially, in the last row of

the table, we have highlighted the distinctive information that each emerging sensor can provide, while conventional sensors failed to do so. For example, their sub-features serve as better indicators for detecting hazardous scenarios compared to those of conventional sensors. Additionally, we have also proposed future research opportunities based on the rather sparse or entirely empty boxes in this table. The details of these discussions can be found in sub-section 2.5.

Remark (4) Figures are quite high level and schematic, lacking technical specificity. Additional technical depth would strengthen the overall quality of the manuscript.

Thank you for your insightful suggestions. In the current manuscript, the number of figures has been reduced to 2 (referred to as *Figure 1* and *Figure 2* in the manuscript).

Figure 1, on page 6, presents the concept of a multi-sensor system for lithium-ion batteries, with examples of using acoustic/ultrasonic, and mechanical sensors to detect the lithium plating. The mechanisms of lithium plating and the schematic experimental configuration of these two types of sensors are also included for clarity. The explanations of this figure can be found mainly in Section 2.

Figure 2, on page 23, shows the developed multi-sensor systems (sensors and hardware) from three different case studies, which are discussed in Section 4.1 - *Sensor development and assembling at early TRL stages*. These examples explored different combinations of conventional and emerging sensors, building the ground for future proof-of-concept studies on multi-sensor system development.

Reply to Reviewer 3

General remarks) The manuscript presents a comprehensive review of using emerging sensor technologies and physics-guided methods for vehicle battery state monitoring. The manuscript reviewed previous efforts in developing state monitoring technologies and emphasizes smart sensors as a future development direction. However, a few problems need to be addressed before the manuscript is published.

We sincerely thank you for going through the manuscript and providing insightful comments.

Remark (1) The author emphasized in the Introduction that "As a result, these battery models are often limited to specific operating ranges." However, the authors did not explain in the paper how emerging sensors based state estimation and fault diagnosis technologies address the "out of distribution" problem.

Thanks a lot for this invaluable comment. Yes, we agree that more evidences are needed to justify the unique benefits of using emerging sensors. In response, we have made the following revisions in Sections 1 and 2.

In Section 1 - *Introduction*, from the end of page 2 to the beginning of page 4, we have stressed the research challenges in degradation and potential safety issues of lithium-ion batteries. Lithium plating has been taken as one example here, explaining its causes, consequences, and limitations of traditional methods (e.g., coulombic efficiency and voltage plateaus analysis). The literature review here lays the ground for introducing emerging sensors as potential solutions to tackle these challenges.

In Section 2 - *Emerging Sensing Techniques in Battery Monitoring*, we have enriched the literature review of each sensor type. The 'out of distribution' problems, including in-suit detection of lithium plating, swelling, and abuses studies have been more focused on.

For example, as mentioned in Section 1, detecting lithium plating in real-time remains challenging with conventional methods. When irreversible lithium plating occurs, either locally or globally, it can cause changes in the battery material density, stiffness, cell thickness, etc. These changes, in turn, affect the propagation speed of sound waves [7] and induce expansion overshoots [8]. Acoustic, ultrasonic, and mechanical sensors can capture these effects through distinct features, such as endpoint differences [7] and expansion ratios [8]. To make these discussions more illustrative, we have included a new figure (referred as to

Figure 1 in the manuscript) that explains the mechanisms of lithium plating, experimental configurations for the two sensor types, and examples of how they can detect plating.

Furthermore, for clarity and better understanding, *Table 1* on page 15 has summarized the discussed literature, classified by sensor types and objectives. From this table, we notice that one of the most significant advantages of using emerging sensors is their ability to elucidate the mechanisms behind various side reactions, particularly those related to safety. Especially, in the last row of the table, we have highlighted the distinctive information that each emerging sensor can provide, while conventional sensors failed to do so. For example, their sub-features serve as better indicators for detecting hazardous scenarios compared to those of conventional sensors. Additionally, we have also proposed future research opportunities based on the rather sparse or entirely empty boxes in this table. The details of these discussions can be found in sub-section 2.5.

Remark (2) Some parts of the manuscript contain well-known information from the literature, which could be reduced to focus more on critical analysis and new insights.

We acknowledge this concern and agree that a better balance is needed. As mentioned in the changes of outlines, we have removed the previous Section 2 to reduce the contents on state estimation with conventional sensors.

Additionally, in the current manuscript, sub-sections 2.1 - 2.4, 3.1, and 4.1 have included more literature review on how each emerging sensor contributes to addressing 'out of distribution' problems, the recent advances in using physics-guided machine learning to enhance the modeling performances, and the proof-of-concept studies of the multi-sensor system. Sub-sections 2.5, 3.2, and 4.2 have been included to discuss the prospective aspects of emerging sensors to offer a deeper view into emerging sensor topics and potential research opportunities.

Lastly, previous figures and boxes have been reduced to only two figures and one table, offering more technical specificity.

- *Figure 1* on page 6, as mentioned in the last answer, presents the concept of a multi-sensor system for lithium-ion batteries, with examples of using acoustic/ultrasonic, and mechanical sensors to detect the lithium plating. The mechanisms of lithium plating and the schematic experimental configuration of these two types of sensors are also included for clarity. The explanations of this figure can be found mainly in Section 2.
- *Figure 2*, on page 23, shows the developed multi-sensor systems (sensors and hardware) from three different case studies, which are discussed in Section 4.1 - *Sensor development and assembling at early TRL stages*. These

examples explored different combinations of conventional and emerging sensors, building the ground for future proof-of-concept studies on multi-sensor system development.

- *Table 1* on page 15, as mentioned in the last answer, has summarized the discussed literature, classified by sensor types and objectives. Based on this table, the main benefits of employing emerging sensors and some future search ideas have been discussed in sub-section 2.5.

We hope the updated contents have brought more critical analysis and new insights.

Remark (3) State monitoring techniques based on emerging sensors share similar workflows with traditional sensors, including threshold selection, feature extraction, parameter identification, and building regression models. It is suggested that the authors make a more detailed comparison between these two technologies and explain why state monitoring based on advanced sensors is poised to become a crucial avenue for future developments.

We appreciate your valuable suggestions. Indeed, the workflows of emerging sensors are similar to those of conventional sensors. In the current manuscript, the application areas of emerging sensors vary from battery characterization, state regression and estimation, multi-physics modeling, degradation mechanisms study, and non-destructive diagnosis to abuse studies, as we summarized and compared in *Table 1* and discussed throughout *Section 2*. Though many studies have been done, only very few studies have offered quantitative comparisons between emerging sensors and conventional sensors.

For example, for state estimation, Bombik et al. [9] extracted features from the ultrasonic sensors and directly used them as static states in an Extended Kalman Filter (EKF) to estimate battery SoC and SoH simultaneously. The validation results show that the proposed method can converge from poorly defined initial states and track well within five cycles, whereas the conventional approach using only voltage measurements fails without any sign of converging.

More quantitative comparisons are available in abuse studies. For example, acoustic/ultrasonic sensors could provide a non-repeatable earlier indicator (than voltage signals) for over-charge [10], a longer time margin (than the temperature-based method) for over-charge induced failure prevention [11]. Similarly, mechanical sensors offer earlier indicators (than temperature signals) for internal short circuit triggered thermal runaway [12] and mechanical abuse [13]. Gas sensors can also supply earlier indicators (than voltage signals) for nail penetration triggered venting [14].

Furthermore, the unique information provided by each sensor type has also contributed to the multi-physics modeling. For example, based on the sound

propagation theories and the layer structures of batteries, Huang et al., [15] developed a pivotal theoretical model to perform layer-resolved characterization of lithium-ion batteries. Similarly, with gas sensors, Wang et al. [5] developed a multi-physics multi-phase model to simulate the venting propagation process of the pack and analysis the byproducts of the venting process. The temperature evolution during thermal runaway propagation and gas concentration after venting agreed well with the experiment measurements, justifying the model fidelity and offering insights into the optimal design of the battery thermal management system. Though these studies do not offer direct quantitative comparisons with conventional sensors, they provide unique insights that conventional sensors cannot, as summarized in the last row of *Table 1*.

It is also important to emphasize that the full benefits of emerging sensors are often realized when combined with conventional sensors, as we discussed in sub-section 2.5. A multi-sensor system is crucial for capturing the complex electro-mechanical-thermal dynamics within lithium-ion batteries. For example, Knehr et al. [16] used the EIS and ultrasonic sensors to study the 'break-in' period between cell formation and steady states. Likewise, Laufen et al. [17] used voltage, strain, and impedance measurements to determine the optimal pressure for enhancing cell cycle lifetime. The case studies in sub-section 4.1 further demonstrate how conventional and emerging sensors complement each other for optimal battery monitor and control.

In summary, while emerging sensors offer promising capabilities, their application remains somewhat scenario-specific. However, as we discussed in sub-sections 2.5, 3.2, and 4.2, the potential for single-sensor use and multi-sensor systems is presented and warrants further exploration.

Remark (4) The value of optical fiber sensors includes simultaneously monitoring the mechanical performance and temperature distribution of lithium-ion batteries. The installation of optical fiber sensors (OFS) is a crucial aspect of fiber-based state monitoring technologies. I suggest that the authors provide a more detailed discussion of these aspects.

Thank you for bringing this point up. Yes, optical fiber sensors have shown great potential, and the internal sensor installation is of great importance.

In sub-section 2.3, studies by Bae et al. [18], Raghavan et al. [6], and Li et al. [19] have been reviewed, as they all injected the fiber Bragg grating (FBG) sensors into the batteries to monitor their dynamics. Furthermore, the requirements for installing sensors into a single cell have been discussed in sub-section 4.1, from the end of page 22 to the beginning of page 23; meanwhile, in sub-section 4.2, we also considered the challenges if one wants to manufacture these internal sensors together with batteries in a large volume.

Remark (5) I suggest that the authors supplement the content related to temperature sensors in Chapter Four. Additionally, Electrochemical Impedance Spectroscopy (EIS) is a mature electrochemical analysis method with commercial implementation solutions suitable for BMS, such as commercial chips and Control Units. EIS can also capture valuable electrochemical information within the battery. Therefore, it is recommended that the authors include additional content related to EIS.

Thanks a lot for the suggestions. In sub-section 4.1 and *Figure 2*, the temperature sensors, both external and internal, have been used in the case studies. When it is considered for external usage, the practical requirements and experiences for external sensor assembling (for example, the location, number, cost, etc.) have been considered. When it is considered for internal usage, the requirements for sensor materials, sensor integration, and potentially the production processes, have also been discussed.

The impedance sensors, as we mentioned in previous answers, have been used together with other emerging sensors. These detailed discussions could be found in Section 2. The installation of impedance sensors, similar to assembling external sensors, has been mentioned in sub-section 4.1.

Remark (6) It is recommended that the authors supplement the working principles of these sensors monitoring the internal state of the battery in Chapter Four. For example, elucidate how the internal information within the battery influences the signal variations detected by the sensors and how this information reflects battery degradation.

We sincerely thank you for this valuable recommendation. After adapting the outline of the manuscript, we believe that it would be more appropriate to address this comment in Section 2 rather than in Section 4.

In sub-sections 2.2, 2.3, and 2.4, we have enriched the literature review of each sensor type by beginning with the working principles of the sensors, the signals they detect, the features that can be extracted from these signals, and the specific battery dynamics they monitor. Given the complexity of the interactions between battery dynamics and sensor signals, we have provided a summary of the relevant literature in *Table 1* to aid understanding.

To answer this comment more specifically, we would like again to take the detection of lithium plating as an example to explain the working principles of how ultrasonic and mechanical sensors can help. As mentioned in Section 1, when

irreversible lithium plating occurs, either locally or globally, it can cause changes in the battery material density, stiffness, cell thickness, etc. These changes, in turn, affect the propagation speed of sound waves [7] and induce expansion overshoots [8]. The sub-figures 1(c) and (d) illustrate the schematic configurations of ultrasonic and mechanical sensors. The resulting features, such as endpoint differences [7] and expansion ratios [8] as shown in sub-figures 1(f) and 1(g), serve as indicators of lithium plating. A detailed explanation of such inference could be found in sub-sections 2.2 and 2.3, respectively.

Similar discussions on the use of emerging sensors to detect and model other phenomena, such as battery abuse, SEI growth, and structural defects, are also included in Section 2. The examples provided throughout the section illustrate the versatility and potential of emerging sensors in understanding complex battery dynamics, as well as their crucial role in improving battery safety and performance.

References

- [1] C. Xu, Z. Fan, M. Zhang, P. Wang, H. Wang, C. Jin, Y. Peng, F. Jiang, X. Feng, and M. Ouyang, “A comparative study of the venting gas of lithium-ion batteries during thermal runaway triggered by various methods,” *Cell Reports Physical Science*, vol. 4, no. 12, 2023.
- [2] T. Cai, P. Valecha, V. Tran, B. Engle, A. Stefanopoulou, and J. Siegel, “Detection of li-ion battery failure and venting with carbon dioxide sensors,” *ETransportation*, vol. 7, p. 100100, 2021.
- [3] Y. Peng, H. Wang, C. Jin, W. Huang, F. Zhang, B. Li, W. Ju, C. Xu, X. Feng, and M. Ouyang, “Thermal runaway induced gas hazard for cell-to-pack (ctp) lithium-ion battery pack,” *Journal of Energy Storage*, vol. 72, p. 108324, 2023.
- [4] G. Wang, D. Kong, P. Ping, J. Wen, X. He, H. Zhao, X. He, R. Peng, Y. Zhang, and X. Dai, “Revealing particle venting of lithium-ion batteries during thermal runaway: a multi-scale model toward multiphase process,” *ETransportation*, vol. 16, p. 100237, 2023.
- [5] G. Wang, D. Kong, P. Ping, X. He, H. Lv, H. Zhao, and W. Hong, “Modeling venting behavior of lithium-ion batteries during thermal runaway propagation by coupling cfd and thermal resistance network,” *Applied Energy*, vol. 334, p. 120660, 2023.
- [6] A. Raghavan, P. Kiesel, L. W. Sommer, J. Schwartz, A. Lochbaum, A. Hegyi, A. Schuh, K. Arakaki, B. Saha, A. Ganguli, *et al.*, “Embedded fiber-optic sensing for accurate internal monitoring of cell state in advanced battery management systems part 1: Cell embedding method and performance,” *Journal of Power Sources*, vol. 341, pp. 466–473, 2017.
- [7] C. Bommier, W. Chang, Y. Lu, J. Yeung, G. Davies, R. Mohr, M. Williams, and D. Steingart, “In operando acoustic detection of lithium metal plating in commercial licoo₂/graphite pouch cells,” *Cell Reports Physical Science*, vol. 1, no. 4, p. 100035, 2020.
- [8] F. B. Spingler, W. Wittmann, J. Sturm, B. Rieger, and A. Jossen, “Optimum fast charging of lithium-ion pouch cells based on local volume expansion criteria,” *Journal of Power Sources*, vol. 393, pp. 152–160, 2018.
- [9] A. Bombik, S. Y. S. Ha, M. F. Haider, A. Nasrollahi, and F.-K. Chang, “Li-ion battery health estimation using ultrasonic guided wave data and an extended kalman filter,” in *2021 IEEE Applied Power Electronics Conference and Exposition (APEC)*, pp. 962–966, IEEE, 2021.
- [10] L. Oca, N. Guillet, R. Tessard, and U. Iraola, “Lithium-ion capacitor safety assessment under electrical abuse tests based on ultrasound characterization and cell opening,” *Journal of Energy Storage*, vol. 23, pp. 29–36, 2019.

- [11] Y. Wu, Y. Wang, W. K. Yung, and M. Pecht, “Ultrasonic health monitoring of lithium-ion batteries,” *Electronics*, vol. 8, no. 7, p. 751, 2019.
- [12] T. Cai, A. G. Stefanopoulou, and J. B. Siegel, “Modeling li-ion battery temperature and expansion force during the early stages of thermal runaway triggered by internal shorts,” *Journal of the Electrochemical Society*, vol. 166, no. 12, p. A2431, 2019.
- [13] Y. Li, W. Wang, X.-G. Yang, F. Zuo, S. Liu, and C. Lin, “A smart li-ion battery with self-sensing capabilities for enhanced life and safety,” *Journal of Power Sources*, vol. 546, p. 231705, 2022.
- [14] S. Koch, K. P. Birke, and R. Kuhn, “Fast thermal runaway detection for lithium-ion cells in large scale traction batteries,” *Batteries*, vol. 4, no. 2, p. 16, 2018.
- [15] M. Huang, N. Kirkaldy, Y. Zhao, Y. Patel, F. Cegla, and B. Lan, “Quantitative characterisation of the layered structure within lithium-ion batteries using ultrasonic resonance,” *Journal of Energy Storage*, vol. 50, p. 104585, 2022.
- [16] K. W. Knehr, T. Hodson, C. Bommier, G. Davies, A. Kim, and D. A. Steingart, “Understanding full-cell evolution and non-chemical electrode crosstalk of li-ion batteries,” *Joule*, vol. 2, no. 6, pp. 1146–1159, 2018.
- [17] H. Laufen, S. Berg, J. Engeser, M. Strautmann, A. Koprivc, C. Rahe, E. Figgemeier, and D. U. Sauer, “Correlation between voltage, strain, and impedance as a function of pressure of a nickel-rich nmc lithium-ion pouch cell,” *Advanced Materials Technologies*, vol. 9, no. 8, p. 2301965, 2024.
- [18] C.-J. Bae, A. Manandhar, P. Kiesel, and A. Raghavan, “Monitoring the strain evolution of lithium-ion battery electrodes using an optical fiber bragg grating sensor,” *Energy technology*, vol. 4, no. 7, pp. 851–855, 2016.
- [19] A. Ganguli, B. Saha, A. Raghavan, P. Kiesel, K. Arakaki, A. Schuh, J. Schwartz, A. Hegyi, L. W. Sommer, A. Lochbaum, *et al.*, “Embedded fiber-optic sensing for accurate internal monitoring of cell state in advanced battery management systems part 2: Internal cell signals and utility for state estimation,” *Journal of Power Sources*, vol. 341, pp. 474–482, 2017.